# *Toxoplasma gondii* exploits the host ESCRT machinery for parasite uptake of host cytosolic proteins

Yolanda Rivera-Cuevas[1], Joshua Mayoral[2], Manlio Di Cristina[3], Anna-Lisa E. Lawrence[1], Einar B. Olafsson[1], Romir K. Patel[1], Dishari Thornhill[1], Benjamin S. Waldman[4,5¤], Akira Ono[1], Jonathan Z. Sexton[6], Sebastian Lourido[4,5], Louis M. Weiss[2,7], Vern B. Carruthers[1] *

**1** Department of Microbiology and Immunology, University of Michigan Medical School, Ann Arbor, Michigan, United States of America, **2** Department of Pathology, Albert Einstein College of Medicine, Bronx, New York, New York, United States of America, **3** Department of Chemistry, Biology and Biotechnology, University of Perugia, Perugia, Italy, **4** Whitehead Institute for Biomedical Research, Cambridge, Massachusetts, United States of America, **5** Department of Biology, Massachusetts Institute of Technology, Cambridge, Massachusetts, United States of America, **6** Department of Medicinal Chemistry, College of Pharmacy, Ann Arbor, Michigan, United States of America, **7** Department of Medicine, Albert Einstein College of Medicine, Bronx, New York, New York, United States of America

¤ Current address: Department of Microbiology and Immunology, Stanford University School of Medicine, Stanford, California, United States of America
* vcarruth@umich.edu

**Data Availability Statement:** Mass spectrometry raw files are available on https://chorusproject.org/ at project number 1733.

## Abstract

*Toxoplasma gondii* is a master manipulator capable of effectively siphoning the resources from the host cell for its intracellular subsistence. However, the molecular underpinnings of how the parasite gains resources from its host remain largely unknown. Residing within a non-fusogenic parasitophorous vacuole (PV), the parasite must acquire resources across the limiting membrane of its replicative niche, which is decorated with parasite proteins including those secreted from dense granules. We discovered a role for the host Endosomal Sorting Complex Required for Transport (ESCRT) machinery in host cytosolic protein uptake by *T. gondii* by disrupting host ESCRT function. We identified the transmembrane dense granule protein TgGRA14, which contains motifs homologous to the late domain motifs of HIV-1 Gag, as a candidate for the recruitment of the host ESCRT machinery to the PV membrane. Using an HIV-1 virus-like particle (VLP) release assay, we found that the motif-containing portion of TgGRA14 is sufficient to substitute for HIV-1 Gag late domain to mediate ESCRT-dependent VLP budding. We also show that TgGRA14 is proximal to and interacts with host ESCRT components and other dense granule proteins during infection. Furthermore, analysis of TgGRA14-deficient parasites revealed a marked reduction in ingestion of a host cytosolic protein compared to WT parasites. Thus, we propose a model in which *T. gondii* recruits the host ESCRT machinery to the PV where it can interact with TgGRA14 for the internalization of host cytosolic proteins across the PV membrane (PVM). These findings provide new insight into how *T. gondii* accesses contents of the host cytosol by exploiting a key pathway for vesicular budding and membrane scission.

**Funding:** We gratefully acknowledge the funding support from U.S. National Institutes of Health grants T32AI007528 (Y.R.C.), F31AI152297 (Y.R.C), F31AI36401 (J.M.), T32GM007288 (J.M.), the University of Perugia Fondo Ricerca Di Base 2019 program of the Department of Chemistry, Biology, and Biotechnology (M.D.C.), the U.S. National Institutes of Health grant T32AI007414 (E.B.O.), University of Michigan Life Sciences Fellowship Program (E.B.O.), U.S. National Institutes of Health grants R37AI071727 (A.O.), R01AI158501 (S.L.), and R01DK120623 (J.Z.S.), the University of Michigan Institute for Clinical and Health Research grant UL1TR002240 (J.Z.S.), U.S. National Institutes of Health grant P30DK034933 (J.Z.S.), the Edward Mallinckrodt, Jr. Foundation (S.L), and U.S. National Institutes of Health grants R01AI34753 (L.M.W.) and R01AI120607 (V.B.C). The funders played no role in the study design, data collection and analysis, decision to publish, or preparation of the manuscript.

**Competing interests:** The authors have declared that no competing interests exist.

## Author summary

Intracellular pathogens exploit their host to gain the resources necessary to sustain infection; however, precisely how the intracellular parasite *Toxoplasma gondii* acquires essential nutrients from its host remains poorly understood. Previous work showed that *T. gondii* is capable of internalizing host derived cytosolic proteins and delivering them to its lysosome-like compartment for degradation. However, the mechanism by which the material is trafficked across the membrane delimiting the replicative vacuole in which the parasite resides remained unclear. Here, we report a role for the parasite effector protein TgGRA14 in the recruitment of the host ESCRT machinery and in the uptake of host cytosolic proteins. Important human pathogens have developed strategies for exploiting the host ESCRT machinery for intracellular subsistence. Our study sheds lights on the strategy used by a eukaryotic pathogen in to exploit the host ESCRT machinery for the internalization of resources from its host cell.

## Introduction

Intracellular pathogens have evolved diverse strategies that rely on the exploitation of host factors to ensure intracellular subsistence, replication, and immune evasion. The protozoan parasite *Toxoplasma gondii* is a successful intracellular pathogen that can infect a wide range of nucleated cells while residing in a non-fusogenic replicative vacuole that helps protect it from host intrinsic defenses. This replicative vacuole, known as the parasitophorous vacuole (PV), is remodeled by a subset of specialized secretory proteins called dense granules proteins (GRAs) [1]. Although this compartment also segregates the parasite from the nutrient-rich environment of the host cytosol, *T. gondii* has developed multiple mechanisms for acquiring the resources it needs to sustain infection [2].

*T. gondii* is capable of internalizing host-derived cytosolic proteins [3]. Once endocytosed, the material is trafficked throughout the endolysosomal system of the parasite and delivered to its lysosomal vacuolar-like compartment (VAC) for degradation by the cysteine protease cathepsin L (TgCPL) [4,5]. However, the molecular mechanisms by which this material is trafficked across the PV membrane (PVM) and is further endocytosed by the parasite remained poorly understood.

Several parasite-host interactions involve the recruitment and exploitation of host factors [6] and organelles to the PVM [7–9]. For example, *T. gondii* is known to exploit a subset of the host endosomal sorting complex required for transport (ESCRT) machinery during its invasion of host cells [10]. More recent work has shown that host ESCRT is also recruited to the PVM during parasite replication [11]. This latter recruitment likely occurs via a distinct mechanism since it involves many more components of the ESCRT machinery than that for cell invasion. ESCRT is involved in various cellular processes including formation of the multivesicular body (MVB), plasma membrane repair, cytokinesis, exosome release, and autophagy [12,13]. The machinery is composed of five complexes ESCRT-0, ESCRT-I, ESCRT-II, ESCRT-III, and the Vps4 complex, which sequentially interact with each other to bud vesicles away from the cytosol [14]. Although the localization of ESCRT components at the PVM is consistent with a possible role in facilitating the budding of vesicles into the PV lumen, to date there are no reports of parasite proteins that interact with host ESCRT during parasite replication.

A variety of important human pathogens have developed mechanisms for recruiting host cell ESCRT machinery to ensure an efficient infection and transmission. Intracellular

pathogens that differ in pathogenesis, such as the human immunodeficiency virus-1 (HIV-1) [15–19], dengue virus [20], hepatitis C virus (HCV) [21,22], Ebola virus [23,24], and *Mycobacterium tuberculosis* [25,26] have all evolved mechanisms to interact with the host ESCRT machinery. However, the contribution of ESCRT machinery in the pathogenesis of an intracellular eukaryotic microorganism remains limited to the aforementioned role in *T. gondii* cell invasion.

Herein we show that disruption of the host ESCRT machinery impairs host cytosolic protein uptake in *T. gondii*. Furthermore, we identified TgGRA14, a PVM resident dense granule protein [27] that possesses ESCRT recruitment motifs, as a host ESCRT-interacting protein. The ESCRT recruitment motifs encoded in TgGRA14 are exposed to the host cytosol and are capable of recruiting ESCRT in the context of HIV-1 virus-like particle release. Additionally, replicating TgGRA14-deficient parasites have a markedly lower percentage of internalized host cytosolic protein compared to wildtype, demonstrating that TgGRA14 contributes to ingestion. Nonetheless, recruitment of host ESCRT components to the PV is not completely dependent on TgGRA14 given that the recruitment of the accessory ESCRT protein ALIX is not affected by disruption of TgGRA14. Importantly, recruitment of TSG101 relies on TgGRA14, and specifically involved the PTAP motif encoded in TgGRA14. Collectively, our data supports a model in which *T. gondii* uses TgGRA14 and likely other GRA proteins to usurp the host ESCRT machinery for the vesicular transport of host cytosolic proteins across the PVM.

## Results

### Disrupting the host ESCRT machinery impairs uptake of host cytosolic proteins

The ESCRT machinery is composed of five complexes (ESCRT-0 –ESCRT-III and VPS4) that subsequently interact with one another to form vesicles that bud away from the cytosol (**Fig 1A**). At the last step, the AAA ATPases, VPS4A and VPS4B, hydrolyze ATP to disassemble the machinery at the membrane and facilitate scission [28,29]. Expression of a dominant negative form of VPS4A (VPS4A$^{EQ}$), which is unable to hydrolyze ATP [30], results in the accumulation of the ESCRT-III component CHMP4B at endosomal structures due to failed scission events. We found that whereas overexpression of VPS4A wildtype (VPS4A$^{WT}$) did not cause overt accumulation of VPS4A$^{WT}$ or CHMP4B at the PV (**Fig 1B**), overexpression of VPS4A$^{EQ}$ resulted in marked association of VPS4A and CHMP4B with the PV (**Fig 1C**). That late acting ESCRT components CHMP4B and VPS4A are not observed to the same extent in VPS4A$^{WT}$ is expected since the dynamics of VPS4 appear to be transient, with puncta relating to VPS4B recruitment in MVB lasting for <4 s [31].

To determine whether *T. gondii* relies on the host ESCRT machinery during infection we analyzed replication following disruption of the host ESCRT machinery through the expression of VPS4A$^{EQ}$. Compared to cells expressing exogenous VPS4A$^{WT}$, there was a moderate replication defect in parasites growing in cells expressing VPS4A$^{EQ}$ (**Fig 1D and 1E**). To test if the host ESCRT machinery plays a role in the uptake of host cytosolic proteins by *T. gondii*, host cells were transiently transfected with a cytosolic fluorescent reporter protein (Venus) in addition to either VPS4A$^{WT}$ or VPS4A$^{EQ}$. The transfected cells were infected with parasites lacking the major lysosome-like VAC protease TgCPL (RΔ*cpl*) to reduce the degradation of the internalized Venus and allow visualizing its signal within the parasite. As expected, based on previous studies [3], RΔ*cpl* parasites are capable of internalizing host-derived Venus. Because only 30–40% of the host cells express Venus, the percentage of parasites with ingested Venus is likely an underestimate of the ingestion efficiency. Interestingly, we found that

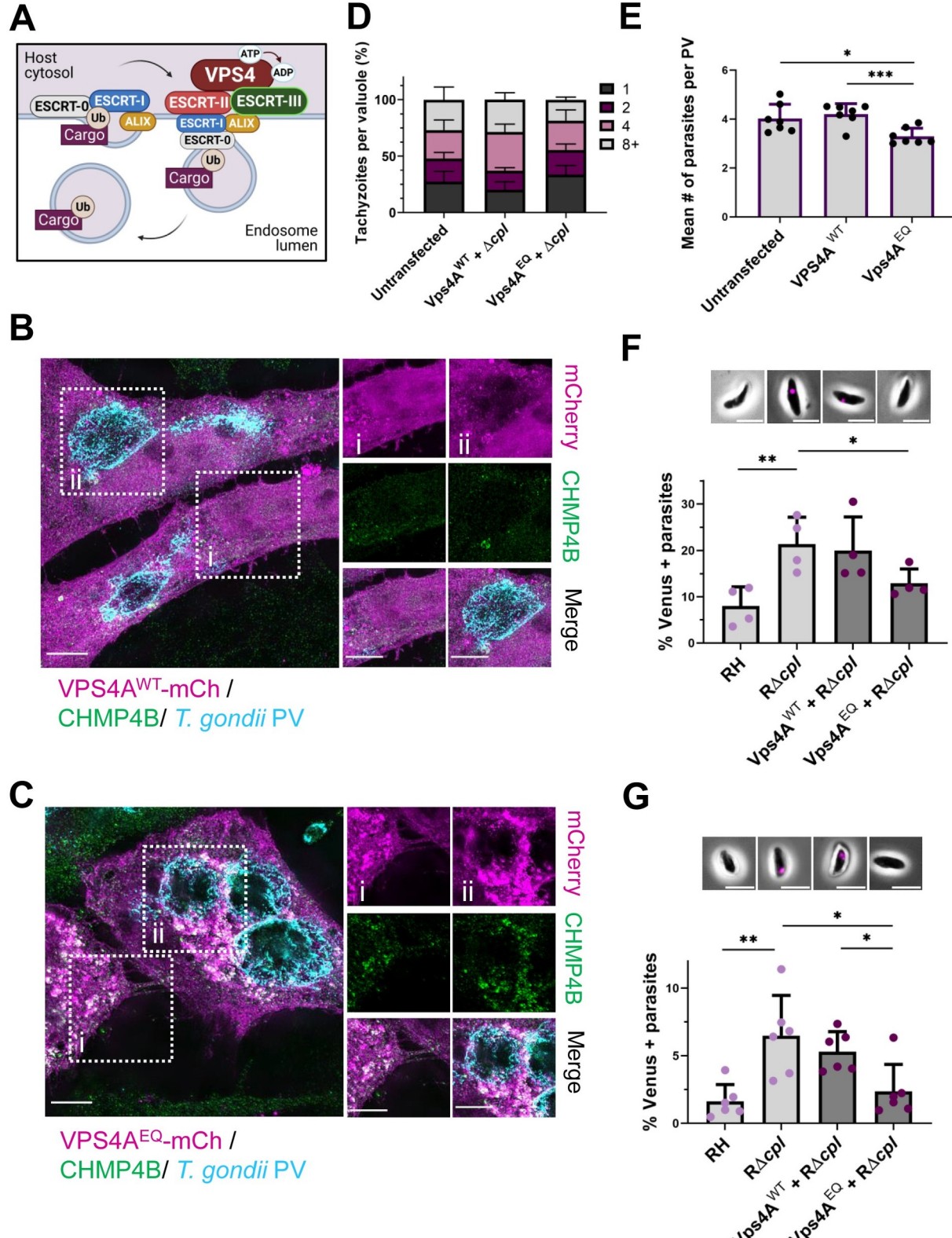

**Fig 1. Disruption of the host ESCRT machinery impairs uptake of host cytosolic proteins by *Toxoplasma gondii*. A.** Schematic of the ESCRT machinery function in MVB formation. Ubiquitinated cargo targeted for degradation is recognized by ESCRT-0 and ECSRT-I, which further recruits ESCRT-II and the ESCRT accessory protein ALIX. At the last steps, ESCRT-III forms spirals at the membrane to mediate

membrane constriction and the VPS4 complex facilitates scission and disassembly of the machinery by hydrolyzing ATP. **B.** Distribution of the ESCRT-III component CHMP4B at the host cytosol (i) and the PV (ii) in HeLa cells transiently transfected with VPS4A$^{WT}$-mCherry and infected with RH parasites for 24 h. **C.** Distribution of the ESCRT-III component CHMP4B at the host cytosol (i) and the PV (ii) in HeLa cells transiently transfected with VPS4A$^{EQ}$-mCherry and infected with RH parasites for 24 h. The PV was labeled with anti-TgGRA7. Images were analyzed by structural illuminated microscopy (SIM). Scale bar is 5 µm. **D.** Growth assay of RH parasites in HeLa cells transiently transfected with exogenous VPS4A$^{WT}$ or VPS4A$^{EQ}$ compared to untransfected control. The percentage of PV with 1, 2, 4 or 8+ parasites was calculated for each cell subset. At least 20 PV were counted per blinded sample. Data represents means from 7 biological replicates. **E.** Mean number of parasites per PV from growth assay. Statistical analysis was by Student's t-test. **F.** Quantification of ingestion of host cytosolic Venus at 30 mpi in RH or RΔ*cpl* parasites harvest from cells were transiently co-transfected with a plasmid encoding cytosolic Venus fluorescent protein expression and exogenous expression of either VPS4A$^{WT}$ or VPS4A$^{DN}$. Representative images for parasites with ingested host-derived cytosolic Venus (shown in magenta) at 30 mpi included at the top. Scale bar is 5 µm. **G.** Quantification of host cytosolic Venus ingestion at 24 hpi in RH or RΔ*cpl* parasites. At least 200 parasites were analyzed per blinded sample. Representative images for parasites with ingested host-derived cytosolic Venus (magenta) at 24 hpi are included at the top. Scale bar is 5 µm. Data represents the mean from $\geq$ 3 biological replicates. Statistical analysis was by Student's t-test. Only statistical differences are shown. $^{*}p < 0.05$, $^{**}p < 0.01$.

exogenous expression of VPS4A$^{EQ}$, but not VPS4A$^{WT}$, impaired RΔ*cpl* accumulation of Venus in newly invaded (**Fig 1F**) and replicating (**Fig 1G**) parasites. These findings suggest that disruption of host ESCRT limits parasite replication and uptake of host cytosolic proteins.

## Identification of TgGRA14 as a candidate ESCRT-interacting protein

Enveloped viruses including HIV-1 can exploit the host ESCRT machinery for viral budding [32]. In the case of HIV-1, the host ESCRT machinery is recruited to the site of viral budding by the Gag protein through the late-domain motifs, such as PTAP and YPX$_{(n)}$L, which are capable of interacting with the ESCRT-I component Tumor suppressor gene 101 (TSG101) and the ESCRT accessory Programmed cell death 6-interacting protein (PDCD6IP/ALIX, ALIX hereafter), respectively [16,33]. To identify *T. gondii* effector proteins as candidates for the recruitment of the host ESCRT machinery, we performed a bioinformatic search using the vEuPathDB *Toxoplasma* Informatics Resources Database (ToxoDB). The first step in our research strategy consisted of a motif search for proteins within the *T. gondii* genome encoding the late domain motif P(S/T)AP. As a second step we looked for proteins that also encode a signal peptide to center our search on secreted proteins. Through this search strategy we identified 33 secretory proteins with a putative P(S/T)AP late domain motif (**S1 Table**). The identified candidates included the rhoptry neck protein RON5, which has been shown to interact with the host TSG101 ESCRT-I protein during cell invasion [34]. However, RON5 is only secreted during invasion and is not present at the host-parasite interface during parasite replication. We considered GRA proteins as better candidates since they are secreted continuously during parasite replication [35]. Three of the 33 genes identified were predicted to be GRA proteins based on data from a recent extensive subcellular proteomics study [36] (**S1 Table**).

Among the 33 identified candidates, we prioritized the PV transmembrane protein TgGRA14 because previous work showed that its C-terminus, which encodes a PTAP motif for putative binding to TSG101, is exposed to the host cytosol [27]. Just downstream of the PTAP motif we also identified a YPX$_{(n)}$L motif for the potential recruitment of ALIX (**Fig 2A**). Since HIV-1 Gag uses both a PTAP and a YPX$_{(n)}$L motif to direct virus budding [18], we predicted that, like HIV-1 Gag, TgGRA14 would associate with ESCRT components at the PVM to promote vesicle formation (**Fig 2B**). If so, we further reasoned that the C-terminal domain of TgGRA14 would interact with ESCRT and could be functionally equivalent to the late-domain containing p6 domain of HIV Gag. To test this, we transfected cells with constructs expressing HIV-1 Gag, HIV-1 Gag lacking the p6 late domain (GagΔ*p6*) and HIV-1 Gag with the p6 domain substituted with a portion of the TgGRA14 C-terminal domain (**Fig 2C**). As

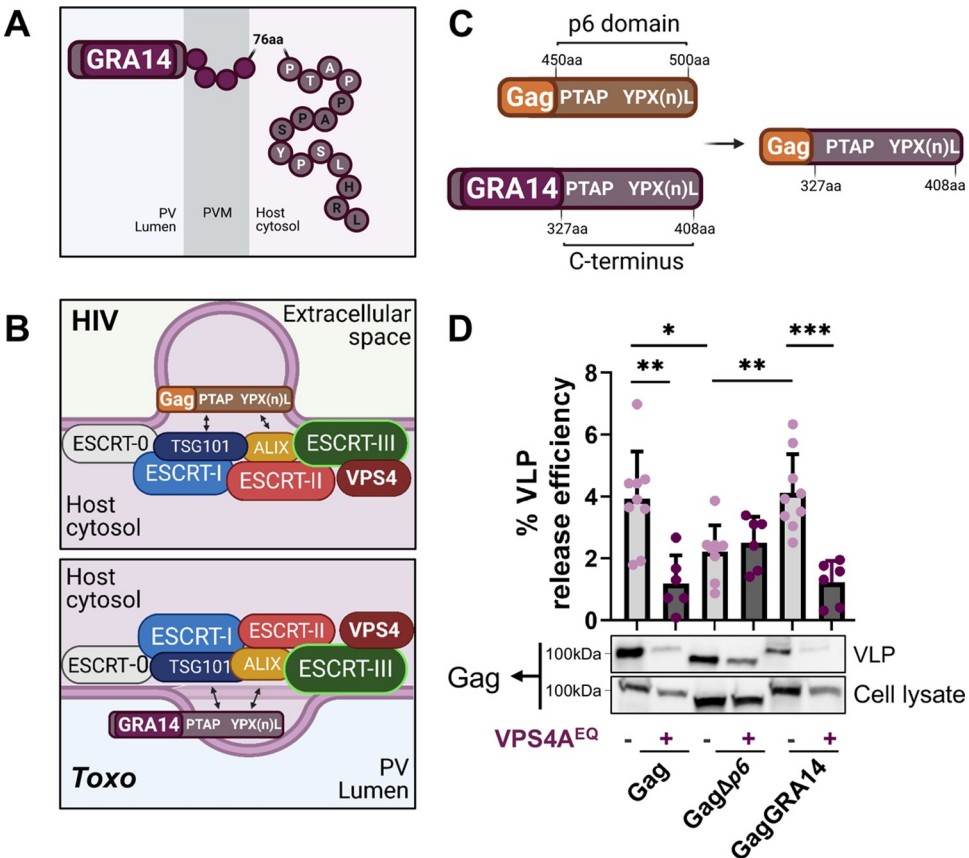

**Fig 2. The GRA14 C-terminal domain facilitates ESCRT-dependent release of VLPs. A.** Schematic of TgGRA14 topology at the PVM. The TgGRA14 C-terminus encoding the late domain motifs PTAP and $YPX_{(n)}L$ is exposed to the host cytosol whereas the N-terminus is exposed to the PV lumen. **B.** Schematic representation of the predicted recruitment of the ESCRT recruitment by TgGRA14 through the late domain motifs in comparison with their known function in HIV-1 budding. The PTAP and $YPX_{(n)}L$ motifs can mediate interactions with host ESCRT-I components TSG101 and the ESCRT accessory protein ALIX, respectively. **C.** Experimental design for the substitution of the HIV-1 Gag p6 domain for the TgGRA14 C-terminus portion encoding late domain motifs to generate the GagGRA14. aa, amino acid. Note that the aa numbering in the hybrid Gag/GRA14 is for GRA14. **D.** Analysis of VLP release by HIV-1 Gag, Gag$\Delta p6$ and GagGRA14. Representative immunoblots of VLP release include top: VLP release sample (100%) and bottom: cellular lysate sample (4%). The role for the ESCRT machinery in GagGRA14 release was assessed by disruption of the ESCRT machinery using the VPS4A dominant negative form (VPS4A$^{EQ}$). Data represents the mean from $\geq 3$ biological replicates. Statistical analysis was by Student's t-test. Only statistical differences are shown. $^*p<0.05$, $^{**}p<0.01$, $^{***}p<0.001$.

expected, expression of HIV-1 Gag sustained the release of virus-like particles (VLPs) in an ESCRT dependent manner, and deletion of the p6 domain diminished VLP production, with residual release being independent of ESCRT (**Fig 2D**). Interestingly, substitution of the late-domain motif portion of TgGRA14 for the HIV-1 p6 domain allowed VLP production that equaled that of native HIV-1 Gag, with release being similarly dependent on ESCRT. Thus, when coupled with HIV-1 Gag, the C-terminal region of TgGRA14 encoding late domain motifs can function for exploitation of host ESCRT for the release of HIV-1 VLPs.

## TgGRA14 is proximal to ALIX and TSG101

Having established that TgGRA14 can exploit the ESCRT machinery in the context of viral budding we next turned our attention to determining if TgGRA14 is linked to host ESCRT during *T. gondii* infection of host cells. First, we stained for ALIX in HeLa cells expressing

endogenously tagged GFP-TSG101[37] to determine if these ESCRT proteins are recruited to the PVM in parasites overexpressing GRA14 C-terminally tagged with HA [27] (R:GRA14$_{OE}$). Analysis by structured illumination microscopy (SIM) showed that ALIX is recruited to the PV (**Fig 3A**), consistent with what has been observed by Cygan, A. M., *et al* [11]. This analysis further revealed that GFP-TSG101 is also abundantly recruited to the PVM where it and ALIX both colocalize with TgGRA14-HA (**Fig 3A**). To further analyze the juxtaposition of ALIX and GFP-TSG101 with TgGRA14, we performed a proximity ligation assay (PLA) for which a positive reaction indicates that two proteins are within 40 nm of one another. This approach has been used previously to study the interactions of ESCRT components in Kaposi's Sarcoma-Associated Herpesvirus (KSHV) infected cells [38,39]. We used the R:GRA14$_{OE}$ parasite strain and probed with anti-HA to detect TgGRA14. Whereas no signal was observed in untagged parasites (WT, RH parasites), we consistently detected a reaction between ALIX and TgGRA14-HA in tagged parasites (**Fig 3B**). As an additional test of specificity, we did not see a reaction between TgGRA14-HA and MAPK7, a host kinase localized throughout the cytosol. We also observed positive reactions between TgGRA14-HA and GFP-TSG101 in infected endogenously tagged GFP-TSG101 HeLa cells that were absent in infected untagged HeLa cells (**Fig 3C**). Together these findings suggest that ALIX and TSG101 are proximal to TgGRA14 in infected cells.

## TgGRA14 influences PV recruitment of GFP-TSG101 but not ALIX

To determine if recruitment of host ESCRT components to the PVM is dependent on TgGRA14 we infected GFP-TSG101 HeLa cells with WT or RΔ*gra14* parasites and probed for host ALIX. After comparing the recruitment of both ESCRT components, we observed defects in the recruitment of GFP-TSG101 but not ALIX in TgGRA14-deficient parasites (**Fig 4A**). Using a spinning disk confocal high content imaging instrument, we quantified the abundance of GFP-TSG101 and ALIX in the PV between WT, RΔ*gra14* and R:GRA14$_{OE}$ to determine if overexpression of TgGRA14 could also affect their recruitment. More specifically, we quantified the ratio of GFP-TSG101 or ALIX associated with the PV (demarcated with anti-GRA1) to that of the cytoplasm (area of the cell occluding the PV and nucleus). This analysis revealed that recruitment of ALIX to the PVM is not affected by overexpression or loss of TgGRA14 (**S1** and **4B** Figs). In contrast, overexpression of TgGRA14 increased recruitment of GFP-TSG101 and ablation of TgGRA14 significantly reduced GFP-TSG101 recruitment to the PVM (**S1** and **4D** Figs). Whether ALIX recruitment is entirely independent of TgGRA14 or through a redundant mechanism in the absence of TgGRA14 remains to be elucidated. Regardless, these findings suggest that TgGRA14 contributes to the PVM recruitment of TSG101.

## TgGRA14 associates with components of host ESCRT

To determine if TgGRA14 physically associates with other ESCRT components, we analyzed interacting proteins by immunoprecipitation (IP) and liquid chromatography tandem mass spectrometry (LC-MS/MS). To avoid potential artifacts due to overexpression, we generated a strain with TgGRA14 endogenously epitope-tagged at the C-terminus with HA in the genetically tractable ME49Δ*ku80* background using CRISPR/Cas9 gene editing (**S2 Fig**). Protein lysates were harvested either from infected human fibroblast cultures (either under tachyzoite (Tz-HFF) or bradyzoite growth conditions (Bz-HFF)) or infected mouse primary cortical neuron cultures (Tz-Neuron). Anti-HA coated magnetic beads were used to enrich for GRA14 protein from lysates, and eluates were harvested for LC-MS/MS. As a control for IP experiments, non-tagged ME49Δ*ku80* parasites were used and processed in parallel to cell cultures

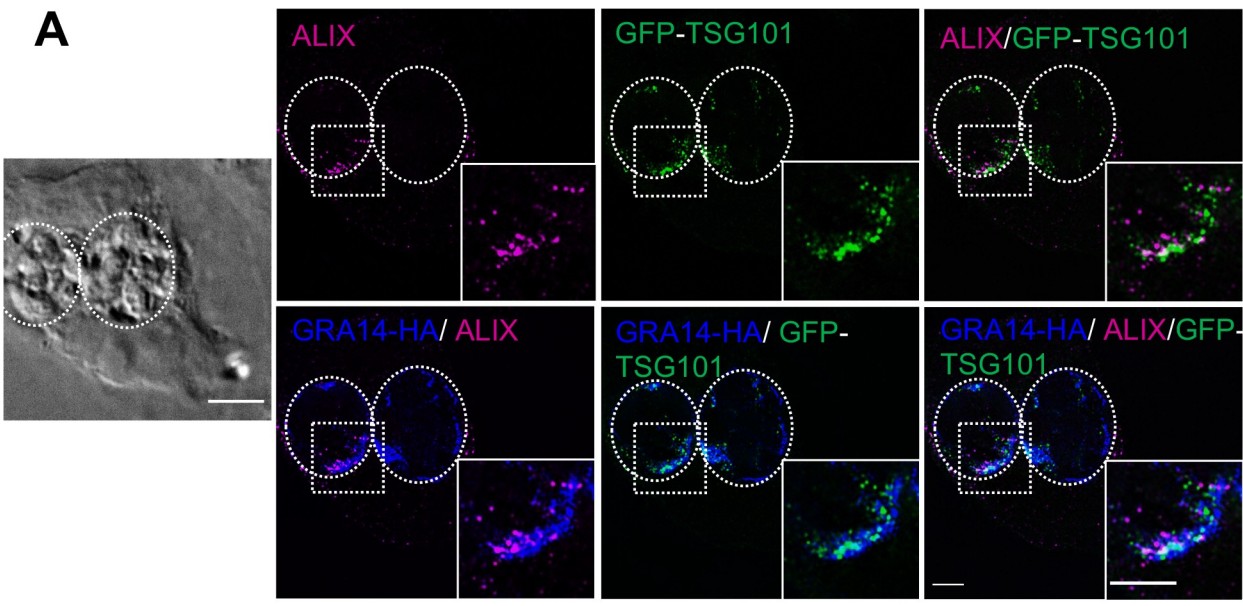

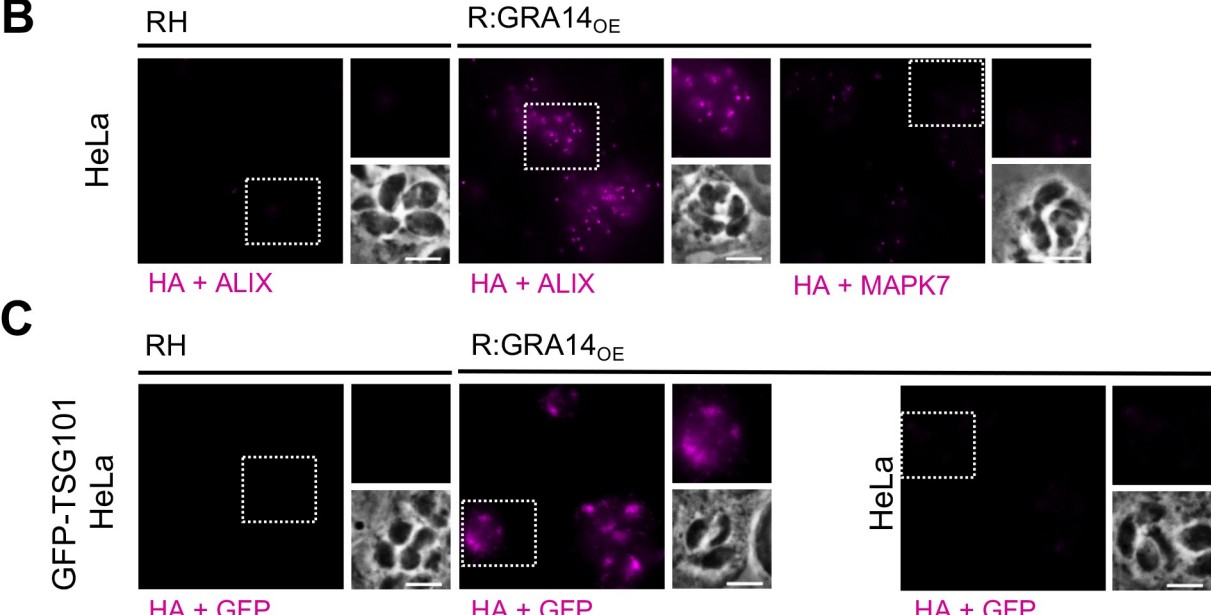

**Fig 3. TgGRA14 is proximal to GFP-TSG101 and ALIX. A.** Recruitment of host ALIX and TSG101 with TgGRA14. GFP-TSG101 HeLa cells infected with TgGRA14-HA over expressing parasites (R:GRA14$_{OE}$) were stained for the anti-ALIX and anti-HA. Representative images analyzed by structured illumination microscopy. 3D Projections. Scale bar is 5 μm. **B.** Proximity Ligation assay (PLA) reaction from samples infected with either WT or R:GRA14$_{OE}$ strains. PLA reactions were performed using antibodies for the host ALIX and the HA-tag to analyze the interaction of TgGRA14 with ALIX. PLA reaction using antibodies against host MAPK7 and HA-tag, was used a negative control. Representative images from at least 3 biological replicates. Scale bar is 5 μm. **C.** PLA reaction from GFP-TSG101 HeLa infected with either WT or TgGRA14-HA strains. PLA reactions were performed using antibodies for GFP and the HA-tag to analyze the interaction of TgGRA14 with TSG101. PLA reaction using antibodies against GFP and the HA-tag in WT HeLa cells was used a negative control. Representative images from at least 3 biological replicates. Scale bar is 5 μm.

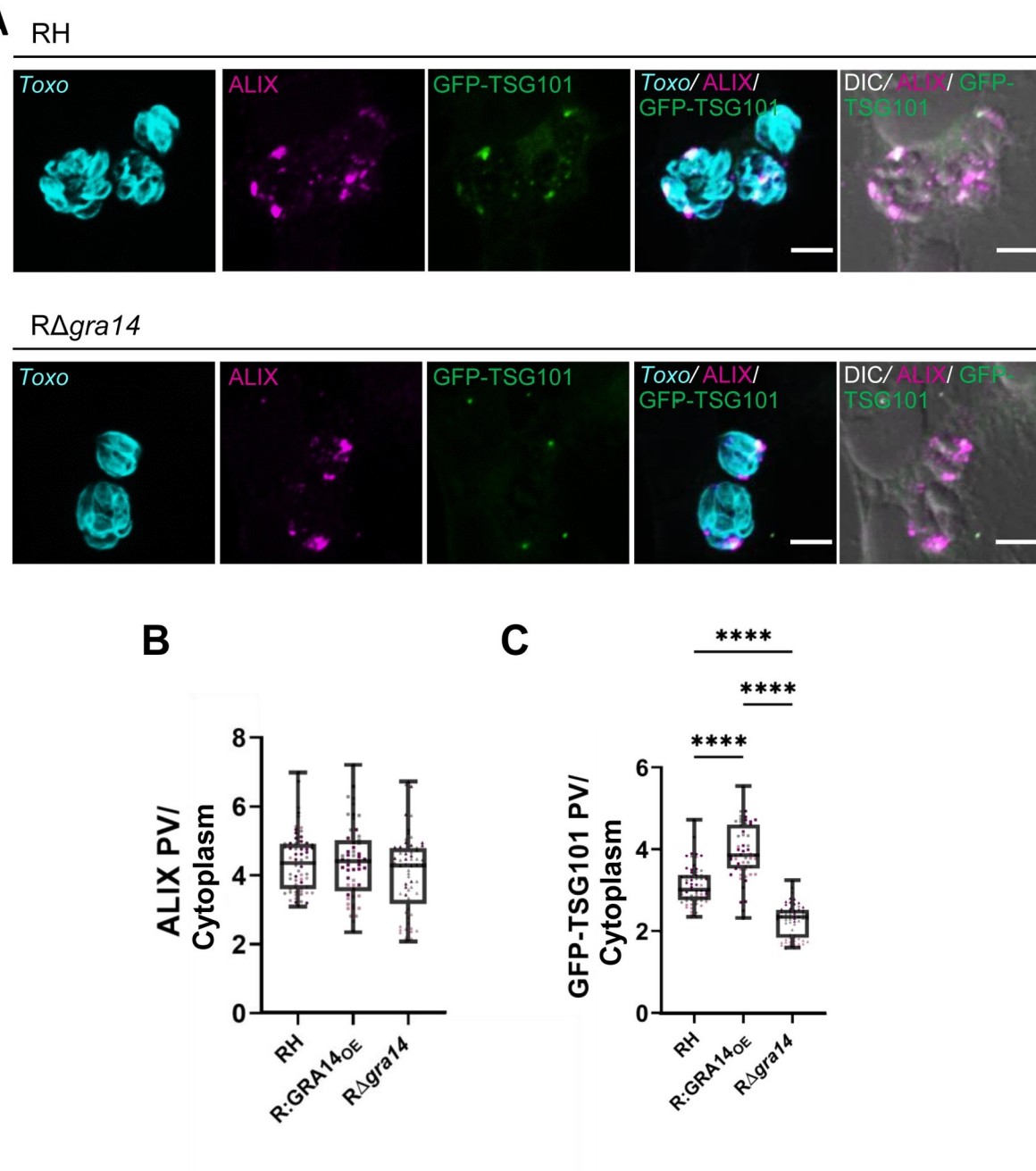

**Fig 4. TgGRA14 influences recruitment of GFP-TSG101 to the PV but not association of ALIX with the PV. A.** Representative images from three biological replicates comparing the recruitment of ALIX and GFP-TSG101 between WT and RΔ*gra14*. Images were analyzed by confocal microscopy. Scale bar is 5 μm. **B.** Quantification of ALIX recruitment to the PV between RH, RΔ*gra14*, R:GRA14$_{OE}$. Data represent the mean ALIX intensity at the PV relative to the cytosol (PV/cytosol). Each point represents the well average. (~18 wells per biological replicates (n = 3)). **C.** Quantification of TSG101 recruitment to the PV between RH, RΔ*gra14*, R:GRA14$_{OE}$ parasites. Data represent the mean TSG101 intensity in the PV relative to the cytosol (PV/cytosol). Each point represents the well average. (~18 wells per biological replicates (n = 3)). Statistical analysis was done using Kruskal-Wallis test. Only statistical differences are shown. ****$p < 0.0001$.

infected with TgGRA14-HA tagged parasites. Notably, results from the IP-MS/MS analysis of tachyzoite infected human fibroblast (Tz-HFF) cultures revealed that most of the significantly enriched host proteins were related to the ESCRT machinery (**Fig 5A and S2 Table**). Several of these interactions were verified by IP/immunoblot (**Fig 5B**). As predicted, both TSG101 and ALIX immunoprecipitated with TgGRA14, along with most components of the ESCRT-I complex, several components of the ESCRT-III complex and the ESCRT-adaptor protein Programed cell death 6 (PDCD6/ALG-2, ALG-2 hereafter) [40,41]. Interestingly, the ALG-2 interacting protein Peflin (PEF1) [42] also immunoprecipitated with TgGRA14; however, a role for PEF1 in ESCRT mediated vesicle formation has not been reported. The IP analysis also

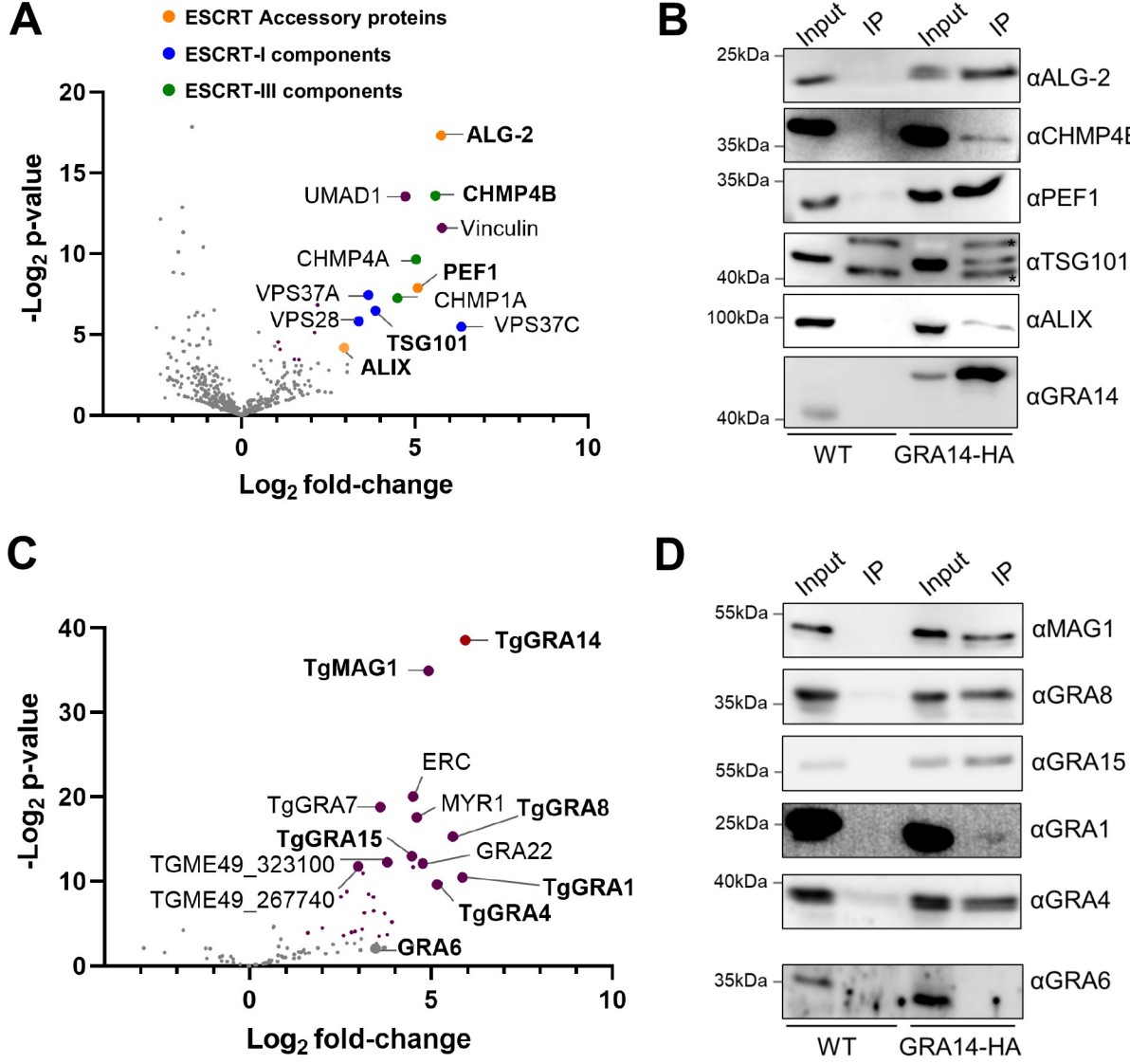

**Fig 5. Host ESCRT components immunoprecipitated with TgGRA14. A.** Volcano plot of the host proteins immunoprecipitated with TgGRA14 under tachyzoite infection conditions showing the ESCRT-accessory proteins (orange), ESCRT-I components (blue), ESCRT-III components (green) and other non-ESCRT associated proteins (purple). Colored dots represent proteins with >1 log2 fold change in tagged vs control lysates and a negative log2 p-value >3.32 (equivalent to p<0.1). **B.** Immunoblots for the analysis of immunoprecipitation samples confirming the TgGRA14 interaction with host ESCRT components. **C.** Volcano plot representative of the *T. gondii* proteins immunoprecipitated with TgGRA14. Some of the enriched dense granule proteins are highlighted in purple. Colored dots represent proteins with >1 log2 fold change in tagged vs control lysates and a negative log2 p-value >3.32 (equivalent to p<0.1). **D.** Immunoblot analysis of immunoprecipitation samples confirming the TgGRA14 interaction with some of the enriched dense granule proteins.

identified TgGRA14 interactions with several other parasite proteins including soluble (TgGRA1, TgMAG1) and transmembrane (TgGRA8, TgGRA15 and TgGRA4) dense granule proteins (**Fig 5C** and **S2 Table**), which were confirmed by IP/immunoblot (**Fig 5D**). Altogether, this data supports a working model in which TgGRA14 interacts with host ESCRT machinery at the PVM where it can potentially trigger vesicle formation to enclose host cytosolic proteins.

## TgGRA14 is important for ingestion of host cytosolic proteins late in infection

To further test the model, we next assessed the role of TgGRA14 in the uptake of host cytosolic proteins using type I RH parasites lacking TgGRA14 (RΔ*gra14*). Parasites were treated with the CPL inhibitor LHVS prior to infection of mCherry inducible HeLa cells and harvested at 4 hours post-infection (hpi) (**S3A Fig**). mCherry inducible HeLa cells have a high percentage of positive cells (>80%) but have lower expression than transiently transfected cells. Since TgGRA14-deficient parasites showed normal ingestion at this time point (**S3B Fig**), we decided to begin treating the parasites with LHVS at 4 hpi to determine if TgGRA14 is necessary for the uptake of host cytosolic proteins later in infection (**Fig 6A**). Quantification of ingestion at 24 hpi in this scheme revealed a significant reduction in uptake of host cytosolic proteins in TgGRA14-deficient parasites (**Fig 6B**). To assess whether this was conserved between strains, we generated TgGRA14-deficient parasites in the type II ME49Δ*ku80* background (MΔ*gra14*) (**S4 Fig**). Like RΔ*gra14*, we observed a significant reduction in the percentage of mCherry-containing parasites in the MΔ*gra14* at 24 hpi (**Fig 6C**). From these data we conclude that TgGRA14 is important for the uptake of host cytosolic proteins late during the later stages of intracellular infection.

## Late domain motifs encoded in TgGRA14 are important for VLP release but not for ingestion

Although HIV-1 Gag encodes several late domain motifs that can interact with multiple ESCRT components to initiate assembly of the machinery, recruitment of the ESCRT

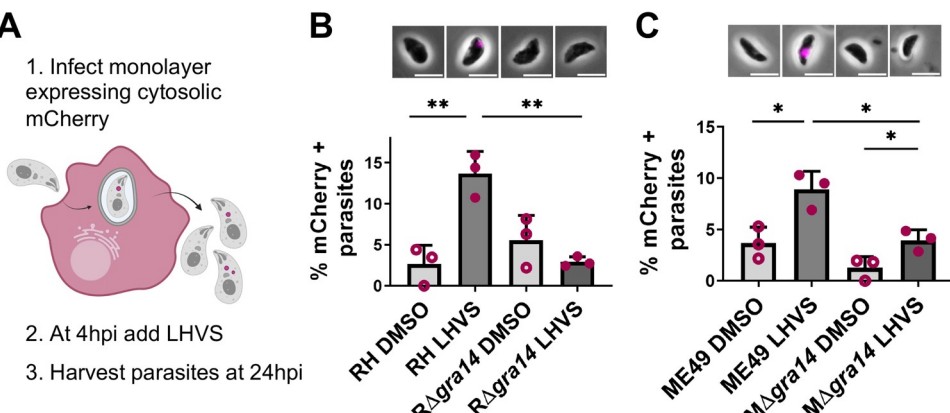

**Fig 6. Replicating parasites deficient of TgGRA14 do not internalize host cytosolic proteins as efficiently as wildtype. A.** Experimental design for the analysis of the internalization of host cytosolic proteins of TgGRA14-deficient parasites. (1) Inducible mCherry HeLa cells were infected with parasites for 4 hours, (2) at 4 hpi, extracellular parasites were removed, and the infected monolayer was treated with LHVS for 20 h, (3) parasites were harvested at 24 hpi and analyzed by microscopy. **B.** Quantification of host cytosolic mCherry uptake at 24 hpi by RH or RΔ*gra14* type I strains treated with DMSO or LHVS for 20 h. **C.** Quantification of host cytosolic mCherry uptake at 24 hpi by ME49 or MΔ*gra14* type II strains treated with DMSO or LHVS for 20 h. At least 200 parasites were analyzed per blinded sample. Data represents the mean from ≥ 3 biological replicates. Statistical analysis was by Student's t-test. *$p < 0.05$, **$p < 0.01$, ***$p < 0.001$.

machinery at the site of viral budding through its interaction with TSG101 appears to be the most critical [43]. Given that TgGRA14 similarly encodes several late domain motifs, we wanted to identify the main late domain motif facilitating the recruitment of the host ESCRT machinery. To that end, we made alanine substitutions in the TgGRA14 PTAP and $YPX_{(n)}L$ motifs (**Fig 7A**). Late domain motif mutations in the hybrid GagGRA14 protein disrupted the release of HIV-1 VLPs, suggesting that these motifs encoded in TgGRA14 are important in the context of ESCRT-dependent HIV-1 budding process (**Fig 7B**).

Next, to determine if TgGRA14 late domain motifs contribute to the recruitment of ESCRT and uptake of host cytosolic proteins, we complemented the R$\Delta gra14$ strain with TgGRA14-HA encoding late domain motif mutations (**S5 Fig**). To verify that the mutations in TgGRA14 did not affect its topology at the PV, we stained semi-permeabilized infected cells with antibodies against HA to label the TgGRA14 C-terminus and an antibody that recognizes the TgGRA14 N-terminus [44]. Whereas signal for both the TgGRA14 N-terminus and C-terminus (HA) was detected in samples that were fully permeabilized with 0.1% saponin (**S6A Fig**), only signal for the C-terminus was detected following semi-permeabilization with 0.00001% saponin, a concentration of saponin that will permeabilize the host plasma membrane but not the PVM (**S6B Fig**). These results, which are consistent with previous findings for wildtype TgGRA14 [27], confirm that the topology of TgGRA14 is not affected by late domain motif mutations at the C-terminus.

Complementation of R$\Delta gra14$ with wildtype TgGRA14 (R$\Delta gra14GRA14^{WT}$) or TgGRA14 encoding mutations in the $YPX_{(n)}L$ (R$\Delta gra14GRA14^{ALIX-}$) restored recruitment of GFP-TSG101 (**Figs 8A and 8B and S7**). However, complementation of R$\Delta gra14$ with TgGRA14 constructs encoding PTAP mutations (R$\Delta gra14GRA14^{TSG101-}$ and R$\Delta gra14-GRA14^{TSG101-ALIX-}$) were impaired in the recruitment of GFP-TSG101 to the PV (**Figs 8A and 8B and S7A**). These findings are consistent with TgGRA14 interacting with TSG101 through the PTAP motif. Although TgGRA14 deficient parasites do not appear to have a virulence

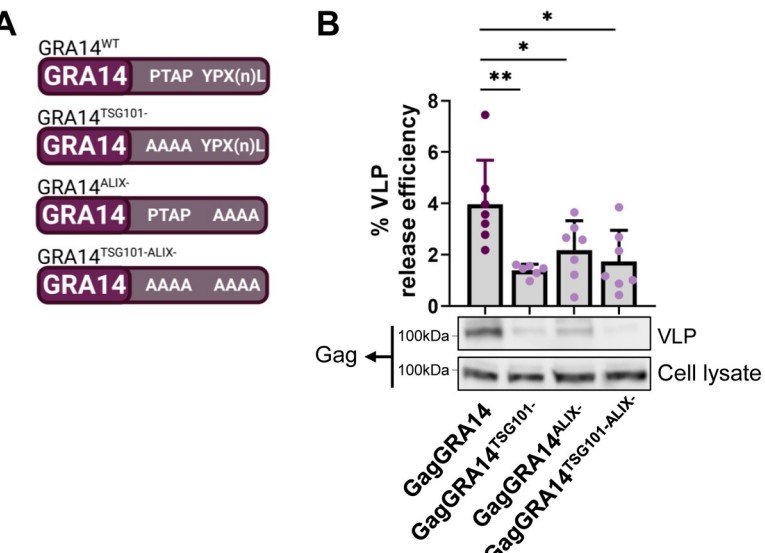

**Fig 7. Late domain motifs encoded in TgGRA14 C-terminus can mediate ESCRT-dependent HIV virus-like particle release. A.** Schematic of the generation of mutants in the late domain motifs PTAP and $YPX_{(n)}L$ encoded by TgGRA14. **B.** Analysis of virus-like particle release by GagGRA14 and GagGRA14 mutants. Representative immunoblots of VLP release include top: VLP release sample (100%) and bottom: cellular lysate sample (4%). Data represents the mean from $\geq 6$ biological replicates. Statistical analysis was by Student's t-test. Only statistical differences are shown. $^*p<0.05$, $^{**}p<0.01$, $^{***}p<0.001$.

defect based on *in vivo* [45–47] or *in vitro* [45,46] data, we decided to measure replication of TgGRA14 late domain mutant strains by quantifying the number of parasites per PV. We also measured the PV size as another approach to assess replication, a strategy previously used [48]. Consistent with the literature, there was no difference in growth between RH,RΔ*gra14*, and the TgGRA14 complementation strains (**S6B and S6C Fig**).

Unexpectedly, however, parasites expressing TgGRA14 late domain motif mutations showed normal internalization of host cytosolic proteins (**Fig 8C**). Taken together, these findings suggest that TgGRA14 late domain motifs are necessary for functional interaction with host ESCRT in the context of viral budding and GFP-TSG101 recruitment, but not for parasite ingestion of host cytosolic proteins.

## Discussion

Herein we show that host ESCRT and TgGRA14 are necessary for parasite ingestion of host cytosolic protein, that TgGRA14 has motifs that support ESCRT-dependent budding of HIV-1 VLPs, and that TgGRA14 co-immunoprecipitates with components of host ESCRT and other parasite secretory proteins in the setting of different parasite growth conditions (e.g., tachyzoite vs. bradyzoite) and host cell environments (e.g., human fibroblasts vs. mouse neurons). We also establish a contribution of TgGRA14 to the PVM recruitment of the ESCRT-I protein TSG101 in a manner dependent upon a PTAP motif in the cytosolically exposed C-terminal domain of TgGRA14. Further, our findings suggest that expression of TgGRA14 is not necessary for PVM recruitment of ALIX despite TgGRA14 having a putative ALIX interaction motif. Finally, we found that although GRA14 is necessary for parasite uptake of host protein, its TSG101 and ALIX motifs are dispensable for this process.

Whereas ESCRT and TgGRA14 are both required for the ingestion pathway late during infection, parasite ingestion of host cytosolic protein soon after invasion depends on ESCRT, but not TgGRA14. This suggests that early in infection *T. gondii* interacts with the host ESCRT machinery through a different mechanism. Although it has been shown that GRA proteins are secreted into the PV within 30 min post-invasion [35], it is unclear how long it takes them to reach their correct destinations and become functional thereafter. By contrast at least some proteins discharged from the parasite rhoptries during invasion are immediately functional since several are required for entry including those released from the rhoptry neck (RON proteins). Indeed, prior work showed that RON4 and RON5 interact with host ALIX and TSG101, respectively [10]. The function of TSG101 and ALIX during *T. gondii* invasion appears to be mainly as adaptor proteins for interaction with the host cytoskeleton and is likely independent of their roles in vesicle budding [10]. However, RON proteins remain associated with a discrete region of the nascent PVM for some time after invasion [49], raising the possibility that they could function in ESCRT-dependent parasite uptake of host cytosolic protein. Additional studies are necessary to test this possibility.

The abundant strategies used by pathogens to exploit the host ESCRT machinery highlight the importance of this host pathway in microbial pathogenesis. The multiple functions of the machinery in the cells provide numerous possibilities for an intracellular pathogen to benefit from it. This is not limited to mammalian host cells and their pathogens, since ESCRT-III and VPS4A homologs can be found in archaea *Sulfolobus spp.*[50,51] and they are exploited by the archaeal *Sulfolobus* turreted icosahedral virus (STIV) for replication [52]. The best described examples of a pathogen usurping the ESCRT machinery are seen in viral egress from the host cell, this is the case for multiple enveloped viruses such as HIV-1[18], the equine infectious anemia virus (EIAV) [43], and the Ebola virus [23,24]. Although a role for the ESCRT-III homologs has not been defined, a similar observation of ring-like structures at budding sites

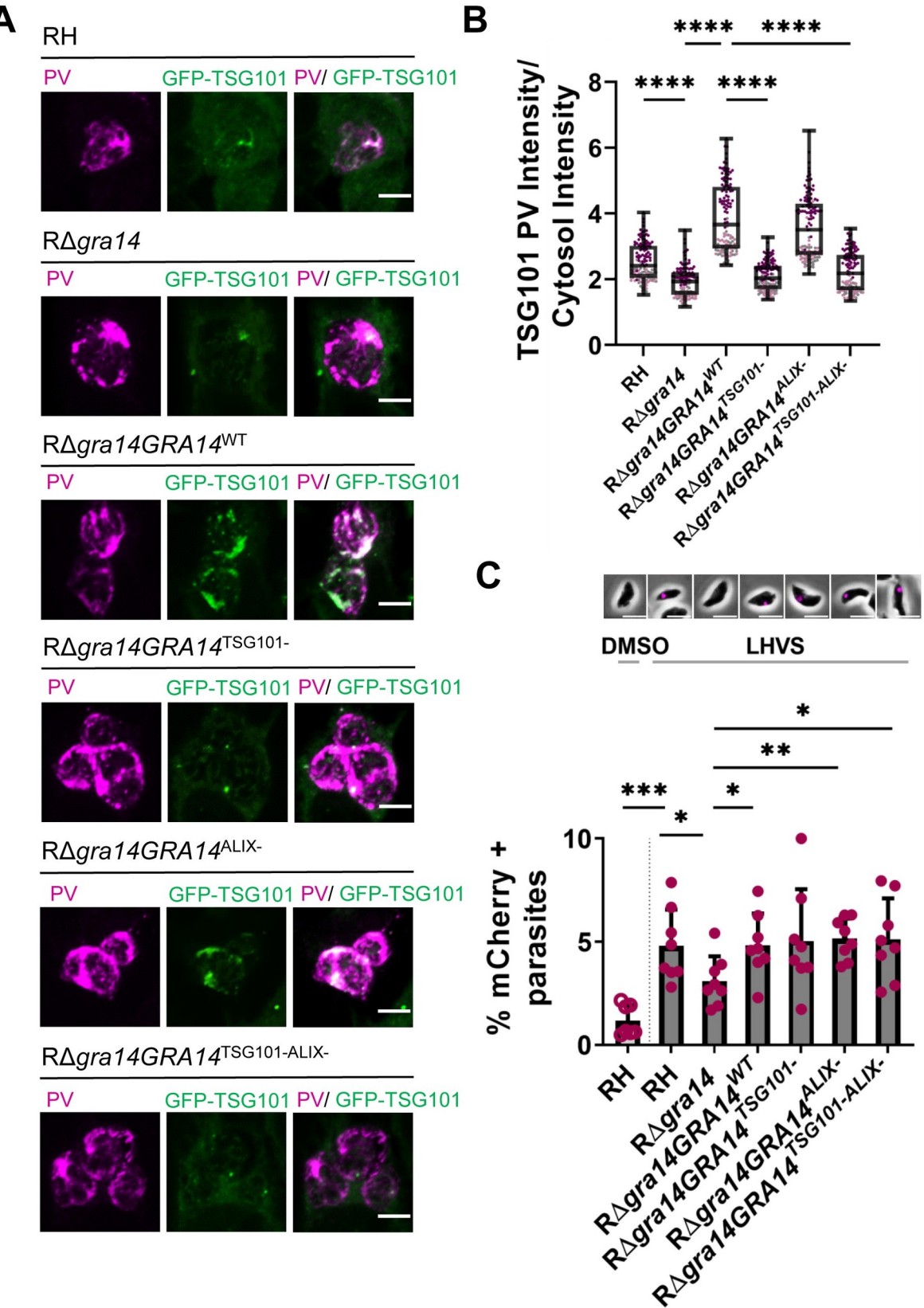

**Fig 8. TgGRA14 PTAP late domain motif is necessary for GFP-TSG101 recruitment but not for uptake of host cytosolic proteins. A.** Representative images from 3 biological replicates for the GFP-TSG101 recruitment in R$\Delta$*gra14GRA14* complemented strains. PVM was labeled using an antibody against TgGRA8. Images were analyzed by confocal microscopy. Scale bar is 5 μm. **B.** Quantification of TSG101 recruitment to the PV between WT, R$\Delta$*gra14* and R$\Delta$*gra14* complementation mutants. Data represent the mean TSG101 intensity in the PV relative to the cytosol (PV/cytosol). Each point represents the well average. (~18 wells per biological replicates (n = 3)). Statistical analysis was done using Kruskal-Wallis test. Statistical analysis between complement strains, RH with $\Delta$*gra14* and $\Delta$*gra14* with $\Delta$*gra14*: *GRA14*$^{WT}$ are shown. \*\*\*\**p*<0.0001. **C.** Quantification of host cytosolic mCherry uptake by R$\Delta$*gra14* and R$\Delta$*gra14GRA14* mutants treated with LHVS for 20 h. At least 200 parasites were analyzed per blinded sample. Data represents the mean from $\geq$ 3 biological replicates. Statistical analysis was by Student's t-test.

has been described for the archaeal *Sulfolobus* spindle-shaped virus-1 (SSV1) [53]. The known mechanisms for enveloped virus recruitment of eukaryotic host cell ESCRT machinery in viral budding is through pathogen proteins encoding late domain motifs, PTAP, YPX$_{(n)}$L and PPXY, which mimic motifs present in ESCRT components and are necessary for the assembly of the machinery [16,54]. These motifs are present in the C-terminus of our candidate *T. gondii* ESCRT-interacting protein TgGRA14. Because the most predominant late domain motifs in viral budding are PTAP and YPX$_{(n)}$L, we decided to elucidate their function in TgGRA14.

Our data demonstrates that the C-terminus portion of TgGRA14 is sufficient to substitute for the HIV-1 p6 domain, providing evidence that these late domain motifs encoded in TgGRA14 can recruit the host ESCRT machinery for HIV-1 VLP budding. Mutations in TgGRA14 late domain motifs impaired HIV-1 VLP release and mutations in the PTAP motif impaired GFP-TSG101 recruitment. Nonetheless, mutations in the late domain motifs had no apparent effect in the uptake of host cytosolic proteins suggesting alternative mechanisms for the ESCRT recruitment by TgGRA14 or additional effector proteins that function in concert with or in parallel to TgGRA14 for protein ingestion. It is, however, also possible that our current assay for ingestion is not sensitive enough to detect changes in the dynamics of this process, like slower rates of uptake.

The P(S/T)AP late domain motif described in retroviral exploitation of the host ESCRT machinery and encoded by ESCRT components such as HRS (ESCRT-0) and ALIX is critical for interaction with TSG101[54]. This highlights the role of proline-rich sequences in host ESCRT assembly. Furthermore, ALG-2 interacts with ALIX and PEF1, another protein enriched in our immunoprecipitation data, through proline-rich regions including a PPYPXnYP motif, where X represents any amino acid and n can vary, but is most often 4[55]. Interestingly, TgGRA14 has a putative PPYPXnYP pseudomotif (PPYVPPMYP) in its C-terminus, which could mediate an interaction with the host ALG-2 as an alternative mechanism for TgGRA14 engagement of the host ESCRT machinery. The ESCRT adaptor protein ALG-2 is capable of interacting with both TSG101[56] and ALIX [57,58], and it can also stabilize an interaction between these two proteins in a Ca$^{2+}$-dependent manner [59]. ALG-2 functions as an adaptor protein for the assembly of the ESCRT machinery at sites of membrane injury [41] and it regulates ALIX function in MVB cargo sorting [40]. This host protein is highly enriched in the PV based on our immunoprecipitation data [11] and although its role at this site remains to be elucidated, it could be acting as a bridge to promote the interaction between ESCRT components. The involvement of ALG-2 could suggest a role for Ca$^{2+}$ in the function of the host ESCRT at the PVM. Further analysis of our TgGRA14 immunoprecipitation data looking for interacting parasite effector proteins encoding putative late domain motifs identified TgGRA8 and TgGRA4 as alternative candidates for the recruitment of ALG-2. However, the topology of these dense granule proteins at the PVM, their ability to interact with ESCRT components and function in ingestion remain to be elucidated. Also, additional dense granule proteins that do not encode putative late domain motifs could still be playing a role in ESCRT function at the PVM. This is the case for several intracellular pathogens which manipulate host

ESCRT machinery via mechanisms that appear to be independent of late domain motif mediated interactions. Thus, it is conceivable that other *T. gondii* effector proteins could be interacting with the host ESCRT machinery in conjunction or independently of TgGRA14. This may explain the ALIX recruitment and residual GFP-TSG101 recruitment to the PV in TgGRA14-deficient parasites. Although our findings suggest an interaction between TgGRA14 and ESCRT, evidence of direct interaction is lacking. Thus, we cannot rule out that disruption of TgGRA14 indirectly affects ESCRT-dependent ingestion of host cytosolic protein by affecting other parasite proteins. Future studies should focus on measuring direct interactions and defining if TgGRA14 functions in concert with other dense granule proteins for the recruitment of the host ESCRT machinery. This would help us understand why there is no virulence defect in TgGRA14-deficient parasites [27] and the importance of the uptake of host cytosolic proteins in *T. gondii* pathogenesis.

Since ALIX is still recruited to the PVM of TgGRA14-deficient parasites, it is conceivable that other transmembrane dense granule proteins are contributing to the interaction with the host ESCRT components at the PVM and compensating for the loss of TgGRA14. TgGRA15 is an interesting TgGRA14-interacting protein since it has also been demonstrated by immunoprecipitation to interact with several ESCRT components [60]. Although its role in the recruitment of the ESCRT machinery remains to be further elucidated, the known function of TgGRA15 in type II strains is to activate NF-κB signaling [61], a host immune response that is also moderately stimulated by TgGRA14[47]. The extent to which TgGRA15 or other TgGRA14-interacting dense granule proteins take part in the uptake of host cytosolic proteins remains unknown. The role of TgGRA7 in the context of host cytosolic protein uptake by *T. gondii* has been studied, but it does not seem to be contributing to this pathway. In contrast, TgGRA2, which is critical for the integrity of the intravacuolar network (IVN), is necessary for efficient ingestion [3]. TgGRA14 is present in the IVN but is not necessary for IVN structure [27]. It is possible that IVN disruption in TgGRA2 impairs TgGRA14 distribution at the PVM and IVN, thus resulting in less ingestion. Alternatively, these proteins could be involved in the uptake of host cytosolic proteins through different mechanisms. The emerging picture is that *T. gondii* has the potential to exploit host ESCRT through a variety of means, which could indicate that it has evolved multiple mechanisms because of a critical reliance on host ESCRT.

Blood stage malaria parasites internalize and digest in their food vacuole large amounts of hemoglobin from the cytosol of infected erythrocytes as a critical source of amino acids for their replication. Although hemoglobin laden vesicles derived from the PVM have been seen directly entering malaria parasites through the mouth-like cytostome and other structures [62–64], little is known about how such vesicles are generated. Components of the ESCRT machinery have been identified in the proteome of mature erythrocytes [65] and in exosomes derived from immature erythrocytes (reticulocytes) [66] Although this raises the possibility that intraerythrocytic malaria parasites similarly exploit host ESCRT for protein ingestion, there is currently no evidence that ESCRT is functional for vesicle biogenesis in erythrocytes. Hepatocytes, in which liver stage malaria parasites replicate, have functional ESCRT [67–69]. However, the extent to which liver stage parasites internalize host protein remains to be determined. Recent work has identified a role for malaria homologs of ESCRT in the generation of extracellular vesicles derived from infected erythrocytes [70]. This study, however, did not distinguish whether such parasite ESCRT homologs function within the parasite, at the PVM, in cytoplasmic membranous structures, or at the erythrocyte plasma membrane. Homology searches fail to detect TgGRA14 homologs in other apicomplexan parasites except for some other tissue cyst-forming coccidian parasites (*Hammondia hammondi*, *Neospora caninum*, and *Besnoitia besnoiti*), but this could be due to the rapid evolution of proteins that reside at

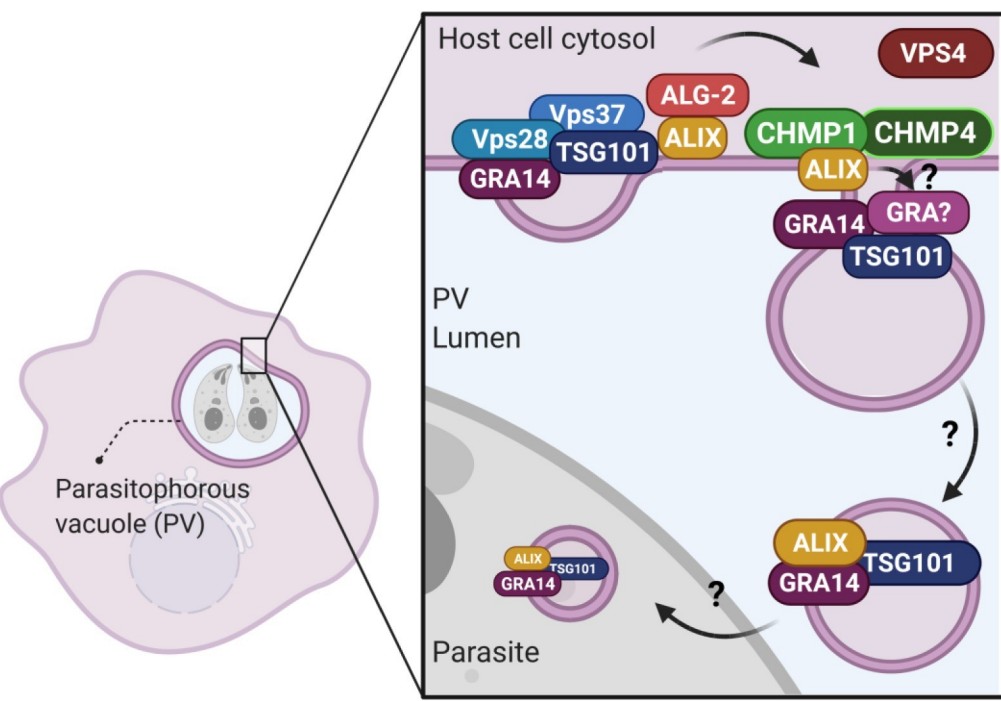

**Fig 9. Model of TgGRA14-ESCRT interactions for the uptake of host cytosolic proteins.** Our data supports a model for the interaction of TgGRA14 with the host ESCRT machinery at the PVM for the uptake of host cytosolic proteins. We hypothesize that through this interaction, vesicles packaging host cytosolic proteins are formed at the PVM and are further endocytosed by the parasites by an unknown mechanism. ESCRT components like TSG101 and ALIX could potentially be packaged within these vesicles resembling what occurs in ESCRT-dependent exosome formation. Furthermore, other transmembrane or soluble dense granule proteins are likely contributing to this pathway.

the parasite-host interface [71]. Attention should be given to future studies dissecting the commonalities and distinctions of host protein uptake by apicomplexan parasites.

Based on our findings, we propose a working model in which *T. gondii* acquires host cytosolic proteins by vesicular trafficking across the PVM happening through the interaction of the host ESCRT machinery with TgGRA14 (**Fig 9**). This model posits that TgGRA14 functions as an ESCRT-adaptor for the recruitment of ESCRT-I components that subsequently interact with the accessory ESCRT protein ALIX, involved in bridging the ESCRT-I and ESCRT-III complexes. Although our work exposes the first mechanistic insight into how *T. gondii* recruits host ESCRT to the PVM during intracellular replication, much remains to be learned about the role of TgGRA14 and other parasite secretory proteins in this process.

## Materials and methods

### Ethics statement

Mouse husbandry and primary neuron dissections were conducted according to guidelines from the United States Public Health Service Policy on Humane Care and Use of Laboratory Animals. Animals were maintained in an AAALAC-approved facility, and all protocols were approved by the Institutional Care Committee of the Albert Einstein College of Medicine, Bronx, NY (Animal Protocol 20180602; Animal Welfare Assurance no. A3312-01).

## Host cell and parasite culture

*Toxoplasma gondii* cultures were maintained in human foreskin fibroblasts (HFF). RHΔ*gra14* and RH GRA14-HA (R:GRA14$_{OE}$) strains were kindly provided by Dr. Peter Bradley of the University of California in Los Angeles (UCLA). The ME49Δ*ku80* strain has been described previously [72].

HFF, HeLa cells, GFP-TSG101 HeLa and Inducible mCherry HeLa cells were maintained in Dulbecco's modified Eagle's medium (DMEM) supplemented with 10% Cosmic Calf serum, 20 mM HEPES, 5 μg/mL penicillin/ streptomycin and 2 mM L-glutamine. Chinese hamster ovary (CHO) cells were maintained in Ham's F12 supplemented with 10% Cosmic Calf serum, 20 mM HEPES, 5 μg/mL penicillin/streptomycin and 2 mM L-glutamine. For bradyzoite growth conditions, induction media was used at the time of HFF infection (DMEM, 50mM HEPES, pH 8.2, 1% FBS) and cultures kept in an ambient air 37°C incubator. For mouse primary cortical neuron infection, pregnant C57Bl/6 mice (Charles River) were euthanized to harvest brains from E15 mouse embryos, from which cortical neurons were dissected. Culture dishes (15cm) coated with poly-L-lysine were plated with 6x10$^6$ neurons each and cultures maintained in Neurobasal media (Thermo Fisher) supplemented with GlutaMAX (Thermo Fisher) and B-27 (Gibco). Cytarabine was added to cultures on DIV4 to minimize non-neuronal contamination. Cultures were maintained up to DIV16 before infection with parasites.

## Bioinformatic search strategy

To identify *T. gondii* proteins encoding late domain motif predicted to interact with the host TSG101 ESCRT-I protein, we performed a search in the *Toxoplasma* informatics resource ToxoDB. The first step in the search strategy consisted of the identification of genes encoding the P-[S/T]-A-P protein motif pattern and this resulted in an output of 620 genes. As a second step we looked for genes that encoded a signal peptide which resulted in 33 candidate proteins (**S1 Table**). Further analysis of candidate genes using the hyperplexed localization of organelle proteins by isotype tagging (hyperLOPIT) data set [36] allowed us to predict the localization of these proteins.

## Generation of parasite strains

Parasite strains used in this study are listed in **S3 Table**. Insertion of the DNA sequence coding for 6x-HA epitopes before the stop codon of TgGRA14 was obtained by transfecting ME49Δ*ku80*Δ*hxg* with the vector GRA14/gRNA2 and a repair template coding for 6x-HA fused at the 5'- and 3'-end to the 40 base pairs upstream and downstream the TgGRA14 stop codon, respectively. Since the vector GRA14/gRNA2 also carried the bleomycin resistance cassette, 24 hpi post transfection parasites were mechanically extruded from HFF using a 26G needle, filter purified and treated for 4 h with 50 μg/ml of bleomycin at 37°C to enrich for parasites that received the CRISPR/Cas9 vector. Bleomycin-treated parasites were allowed to expand in HFF monolayer until sufficient for cloning into 96-well plates to isolate single clones. Identification of clones carrying the correct integration of the 6x-HA tag was carried out as described previously [73] using primer pair P9 and P10 (**S4 Table**) to detect fusion of the tag to the TgGRA14 end. The ME49Δ*ku80* GRA14-HA$_{6x}$ strain was further validated by immunofluorescence using anti-GRA14 and anti-HA.

To knockout TgGRA14 in a type II strain, CRISPR-Cas 9 gene editing was used to disrupt the gene in the ME49Δ*ku80*Δ*hxg* parasites. ME49Δ*ku80*Δ*hxg* was generated by transfecting ME49Δ*ku80* tachyzoites with 150 μg of a vector expressing Cas9 and two guide RNAs targeting the 5'- (gRNA1 5'- GGCUUGUUGUUUCCGUGCAG -3') and 3'- (gRNA2 5'- ACGUUAGU

AAACUAACGCA -3') ends of the coding region of the *HXGPRT* gene and 10 μg of a repair template. The repair template was obtained by annealing the two complementary 80 bp oligo-nucleotides P15 and P16 (**S4 Table**). composed of the fusion of 40 bp sequences that are homologous to sequences upstream and downstream of the Cas9 cleavages sites in the *HXGPRT* gene, respectively. After transfection, parasites were grown in presence of 6-thiox-anthine (200 μg/ml) for 1 week and then cloned. Deletion of the *HXGPRT* gene was tested by PCR (P17 + P20 and P18 + P19 for the absence of the *HXGPRT* gene and P17 + P18 to assess the size of the genome fragment left between the 2 Cas9 cuts) and sensitivity to mycophenolic acid (MPA; 25 μg/ml) and xanthine (50 μg/ml).

Deletion of *gra14* was achieved by transfecting the strain ME49Δ*ku80*Δ*hxg* with the vectors GRA14/gRNA1 and GRA14/gRNA2, expressing Cas9 and gRNAs (gRNA1: 5'-GCTAGCAGA GGTGAATGTAA-3'; gRNA2: 5'-CCAGAGACCAAGCGAATAGA-3') directing Cas9 nucle-ase to cut the *gra14* gene upstream the start codon and over the stop codon, respectively, along with a repair template carrying the HXGPRT (HXG) resistance cassette flanked at its 5'- and 3'-ends by 40 base pairs homologue to the regions immediately upstream and downstream the two Cas9 cuts, respectively. After selecting for HXG resistant with mycophenolic acid (25 μg/ml) and xanthine (50 μg/ml), parasites were cloned in 96 well-plates and single plaques were screened as previously described [73] using primers to detect the absence of the TgGRA14 gene using primers P3 and P4 (**S4 Table**). The ME49Δ*ku80*Δ*gra14* strain was further validated by immunoblot by probing for GRA14.

RHΔ*gra14* complementation strains were generated by transfecting the parasites with pGRA14-HA^wt^_CAT, pGRA14-HA^TSG101-^_CAT, pGRA14-HA^ALIX-^_CAT or pGRA14-HA^TS-G101-ALIX-^_CAT plasmids (**S5 Table**), linearized using FspAI (Thermo Fisher Scientific, Cat #ER1661). Post-transfection, the parasites were selected with chloramphenicol 20 μM and clones confirmed by PCR using the primer pair P9 and P10, Immunofluorescence and immu-noblot by probing for GRA14 and HA.

## Transient transfection of host cells

To transfect HeLa cells, 4.0 x$10^5$ cells were seeded per well in a 6-well plate. Each well was transfected with 2 μg of DNA using Lipofectamine 2000 transfection reagent (Thermo Fisher Scientific, Cat# 11668027) following the manufacturer's instructions. Cells were either fixed or infected at 16–18 h post-transfection.

## Immunofluorescence and immunoblot

Antibodies used in this study are listed in **S6 Table**. Samples were fixed with 4% methanol-free paraformaldehyde (PFA) in PBS for 10 min, permeabilized with 0.1% Triton X-100 for 10 min and blocked with 10% FBS for 20 min. Primary and secondary antibodies were diluted in 1% FBS prior to 1 h incubations at RT. Images were taken using a Zeiss Axiovert Observer fluorescence microscope, a Nikon X1 Yokogawa spinning disk confocal micro-scope, Nikon Structured Illumination microscope, Nikon X1 Yokogawa Spinning disk con-focal or a Yokogawa CellVoyager 8000 spinning disc confocal high content imaging instrument.

Nitrocellulose membranes were blocked with 5% milk prior to primary antibody incuba-tion and subsequently incubated with horseradish peroxidase (HRP) conjugated secondary antibodies. For detection, the membranes were incubated for 2 min with the SuperSignal West Pico PLUS Chemiluminescent substrate (Thermo Fisher Scientific, Cat# 34580). Chemilumi-nescent signal was visualized using the Syngene PXi6 imaging system.

## Growth assay following ESCRT disruption

HeLa cells were grown on coverslips and transiently transfected with either pVPS4A_WT-mCherry (VPS4A^WT) or pVPS4A_EQ-mCherry (VPS4A^EQ). The monolayer was infected with $1.2 \times 10^6$ RH parasites, then fixed at 24 hpi with 4% PFA. To visualize individual parasites and delineate the PV, the coverslips were stained with rabbit anti-GAP45 and mouse anti-GRA7. The number of parasites per vacuole was quantified for transfected cells, identified by the expression of mCherry, and untransfected cells. Seven biological replicates were collected and at least 20 PVs analyzed per replicate as blinded samples.

## Parasite ingestion assay following ESCRT disruption

HeLa or CHO cells were transiently transfected with 2 µg of pVenus, for the expression of cytosolic fluorescent reporter, and 1 µg of either pVPS4A_WT-mCherry or pVPS4A_EQ-mCherry as described above. Following overnight incubation, the cells were infected with $5.0 \times 10^5$ parasites per well. As a negative control, RH WT strain was used to infect cells expressing only pVenus. To detect internalized host-derived cytosolic material we used the RHΔ*cpl* strain to infect cells expressing pVenus, as a positive control, and cells co-transfected with pVenus and pVPS4A-mCherry isoforms. At 24 hpi parasites were harvested as previously described [3]. Briefly, parasites were harvested on ice by scraping and syringing. The parasites were then treated with a 1 mg/mL pronase-0.01% saponin-PBS solution for 1 h at 12˚C, centrifuged at 1,500 g for 10 min and washed 3 times before adding to Cell-Tak coated slides. The parasites were fixed and permeabilized with 0.1% Triton X-100 prior to imaging.

## Virus-like particle assay

The plasmids pGag_Venus, pRev and pVphu were used for the generation of HIV virus-like particles as previously described [74–76]. To delete the p6 domain (HIV-1 Gag amino acids 450–500) encoding the late domain motifs and generate pGagΔ*p6*_Venus, pGag_Venus was linearized using SwaI and SmaI. To introduce back the encoding region upstream of the p6 domain the encoding region from the 664-base pair (bp) to 1456 bp (a 793 bp fragment-GagInsert) was generated by PCR and introduced into the linearized pGag_Venus by Gibson assembly. For the generation of the pGagGRA14_Venus construct, in addition to the GagInsert, a second insert was made by amplifying TgGRA14 C-terminus from *T. gondii* cDNA (Region encoding amino acids 327–408-GRA14CtermInsert) and both fragments were introduced into the linearized pGag_Venus vector by Gibson Assembly. The plasmids were confirmed by Sanger Sequencing using the primer P8.

HIV-1 Gag virus-like particles (VLPs) were collected as previously described [74]. Briefly, HeLa cells were transfected with pRev, pVphu and pGag_Venus constructs using Lipofectamine 2000. Following overnight incubation, the supernatant containing the released VLPs was filtered and ultracentrifuged at 35,000 rpm for 45 min at 4˚C to collect the VLP pellets which were then lysed with 0.5% Triton X-lysis buffer. The cell lysates were prepared by lysing the monolayer with the same lysis buffer. The lysates (obtained by loading 100% VLP and 4% cell lysates on SDS-PAGE gel) were analyzed by immunoblot using a human anti-Gag. Band intensity was quantified using Image J. Total Gag corresponds to the sum of cell- and VLP-associated Gag. The VLP release efficiency corresponds to the fraction of Gag that was released as VLP relative to the total Gag.

## Generation of plasmids encoding GRA14 late-domain mutants

The predicted late domain motifs encoded in TgGRA14 C-terminus were mutated in the pGag-GRA14_Venus by site-directed mutagenesis (Q5 Site-Directed mutagenesis kit NEB,

Cat#E0554S). Primers encoding mutation alanine substitutions for PTAP or YPXL were used to generate pGagGRA14$^{TSG101-}$_Venus and pGagGRA14$^{ALIX-}$_Venus (P11 + P12 and P13 + P14). The PTAP motif was mutated using primers P11 + P12 with the pGagGRA14$^{ALIX-}$_Venus as a template to generate the double- late domain mutant pGagGRA14$^{TSG101-ALIX-}$_Venus. Mutations were confirmed by Sanger Sequencing using the primer P8.

To generate pGRA14-HA_CAT plasmid encoding late domain motif mutations, a chloramphenicol selectable marker was introduced into the pGRA14-HA plasmid that encodes TgGRA14 C-terminal tagged with a single copy of the HA epitope tag under the TgGRA14 promoter (kindly provided by Dr. Peter Bradley). The pGRA14-HA and pTub_CAT plasmid was digested with ApaI and XbaI. The chloramphenicol selectable marker was gel-purified and was ligated to the vector (linearized pGRA14-HA) following incubation with the T4 DNA ligase. The construct was transformed into DH5α and single colonies analyzed for correct insertion of the selectable marker cassette using NcoI and NotI, two unique restriction enzymes that would generate a 3.5 kb product in pGRA14-HA_CAT and not pGRA14-HA. Late-domain motif mutations in pGRA14-HA_CAT was generated by site directed mutagenesis as described above (P11 + P12 and P13 + P14). The plasmids pGRA14-HA$^{wt}$_CAT, pGRA14-HA$^{TSG101-}$_CAT, pGRA14-HA$^{ALIX-}$_CAT and pGRA14-HA$^{TSG101-ALIX-}$_CAT were confirmed by Sanger Sequencing using primer P8.

## Proximity ligation assay (PLA)

Proximity ligation assay (PLA) was performed using the Duolink In Situ Red Started Kit Mouse/Rabbit (Millipore Sigma, Cat# DUO92101-1KT) following manufacturer's instructions. Briefly, wildtype HeLa or GFP-TSG101 HeLa cells were seeded in coverslips and infected with either RH or RH:GRA14$_{OE}$. At 24 hpi the infected monolayer was fixed with 4% PFA and permeabilized with 0.1% Triton X-100 for 10 min. In a humidity chamber, the samples were blocked using Duolink blocking buffer at 37°C for 1 h then incubated with primary antibody at room temperatures for 1 h. The coverslips were incubated with the PLA secondary antibodies anti-Rabbit PLUS and anti-Mouse MINUS for 1 h at 37°C followed by a ligation step for 30 min and a final amplification step for 100 min, both also at 37°C. Samples of HeLa cells infected with RH strain were used as a negative control.

## TgGRA14-HA immunoprecipitation

HFFs or mouse primary neurons were grown in 15 cm dishes and infected with 3–4.5 x10$^7$ or 1.5–2 x10$^7$ parasites respectively per dish using either ME49Δ*ku80* GRA14-HA$_{6x}$ or the untagged control ME49Δ*ku80*. For bradyzoite infection conditions, 15cm HFF cultures were infected in bradyzoite induction media with 1.5 x10$^7$ parasites per dish. At 48 hpi (or 72hpi for bradyzoite conditions), the samples were washed with cold 1x PBS three times and before scrapping the samples in cold lysis buffer (50 mM Tris pH 7.6, 200 mM NaCl, 1% Triton X-100, 0.5% CHAPS + complete protease inhibitor/Roche Cat# 11836153001). The samples were mechanically disrupted by passing them through a 27.5 syringe five times before sonication on ice (1 sec on, 1 sec off, 20% amplitude, 30 cycles). After a 30 min incubation on ice, the lysed samples were centrifuged at 1,000xg for 10 min at 4°C to remove any remaining intact cells. The supernatant was collected (Input sample) and incubated overnight with anti-HA magnetic beads (Pierce, Cat# 88836). Following incubation, the beads were collected on a magnetic stand for 2 min and washed four times with lysis buffer and four times with wash buffer (50mM Tris pH 7.6, 300mM NaCl, 1% Triton X-100, + complete protease inhibitor/Roche Cat# 11836153001). The beads were resuspended in 1x Laemmli sample buffer, boiled for 5 min, and the immunoprecipitated protein eluted by collecting the supernatant following

incubation in the magnetic stand for 2 min. The collected immunoprecipitated proteins were reduced, alkylated, and digested into peptides with trypsin on S-Trap micro columns (Protifi) per manufacturer instructions.

## LC-MS/MS acquisition and analysis

For peptide samples from all Co-IP experiments, samples were resuspended in 10 μl of water + 0.1% TFA and loaded onto a Dionex RSLC Ultimate 300 (Thermo Scientific, San Jose, CA, USA), coupled online with an Orbitrap Fusion Lumos (Thermo Scientific). The mass spectrometer was set to acquire spectra in a data-dependent acquisition (DDA) mode. Briefly, the full MS scan was set to 300–1200 m/z in the orbitrap with a resolution of 120,000 (at 200 m/z) and an AGC target of 5x10e5. MS/MS was performed in the ion trap using the top speed mode (2 secs), an AGC target of 10e4 and an HCD collision energy of 30. Raw files were searched using Proteome Discoverer software (v2.4, Thermo Scientific) using SEQUEST as search engine. We used either the SwissProt human or mouse databases combined with the Toxoplasma database (Release 44, ME49 proteome obtained from ToxoDB). The search for total proteome included variable modifications of methionine oxidation and N-terminal acetylation, and fixed modification of carbamidomethyl cysteine. Trypsin was specified as the digestive enzyme. Mass tolerance was set to 10 pm for precursor ions and 0.2 Da for product ions. Peptide and protein false discovery rate was set to 1%. For quantitative analysis, peptide intensity values were log2 transformed, normalized by the average value of each sample, and missing values were imputed using a normal distribution 2 standard deviations lower than the mean. Peptide $log_2$ fold changes were then averaged to obtain a single protein $log_2$ fold change. Statistical significance was assessed among the three replicates for each condition using a heteroscedastic T-test (if p-value < 0.05) comparing peptide quantification between test and control samples for each condition.

Mass spectrometry raw files are available on https://chorusproject.org/at project number 1733.

## Parasite ingestion assay using inducible mCherry HeLa cells

Inducible mCherry HeLa cells were generated following the methods for generating the inducible mCherry CHO-K1 cells previously described [4]. Briefly, the cells were transfected with pTRE2-mCherry, for the mCherry expression under a tetracycline-inducible promoter, and pTet-ON, encoding the reverse tet-responsive transcriptional activator. Co-transfected cells were selected with 200 μg/mL hygromycin B and 200 μg/mL geneticin. Resistant cells were tested for their induction of mCherry following doxycycline treatment and were sorted for the brightest mCherry signal. For ingestion experiments, 1.5 x10^5 cells are seeded per well in a 6-well plate. Cytosolic mCherry expression is induced by adding 2 μg/mL doxycycline for 4-days. The cells are then infected with 5.0 x10^5 type I parasites or 1.0 x10^6 type II parasites. For the 4 hpi time point, the parasites were then treated with 5 μM LHVS for 24 h prior to infection of mCherry+ cells. For the 24 hpi time point, the monolayer was washed at 4hpi and fresh media containing 5 μM LHVS is added for 20 h. Parasites were harvested as previously described.

## ALIX and GFP-TSG101 recruitment

HeLa cells expressing endogenously tagged GFP-TSG101[37] were grown on coverslips and infected with R:GRA14$_{OE}$ parasites for 24 h. The samples were fixed and stained for anti-ALIX and anti-HA as described above. Samples were imaged using a Nikon A1R structured illumination microscope.

Initially the recruitment of ALIX in GFP-TSG101 infected cells was analyzed by staining the samples with anti-HA and anti-GRA8 as a marker for the PVM and anti-GAP45 to label the parasites. Samples were imaged using a Nikon X1 Yokogawa Spinning disk confocal. A Z-stack spanning 2 μm with 0.2 μm intervals was captured for each channel were then collapsed into maximum intensity projections.

To compare the recruitment of ALIX between WT, TgGRA14-deficient parasites and a TgGRA14 overexpressing strain, HeLa cells were seeded in a poly-D-lysine coated 384 well plate (4,000 cells per well) 2 h prior to infection with either RH, RΔ*gra14* and R:GRA14$_{OE}$ at a MOI 5 for 24 h. The 384-well plate was fixed and stained for anti-ALIX, anti-GRA1, HCS Cell-Mask Deep Red (CMDR, ThermoFisher) and Hoechst (Sigma). For the analysis for the recruitment of TSG101, GFP-TSG101 HeLa cells were seeded in a 384 well plate 2 h prior to infection with either RH, RΔ*gra14*, R:GRA14$_{OE}$ and RΔ*gra14* complementation strains at a MOI 5 for 24 h. The 384-well plate was fixed and stained for anti-GRA1, HCS CellMask Deep Red (CMDR, ThermoFisher) and Hoechst (Sigma).

Both plates were imaged at 20x using a Yokogawa CellVoyager 8000 spinning disc confocal high content imaging instrument. 9 fields of view (FOV) were captured from each well. A Z-stack spanning 10 μm with 1 μm intervals was captured for each FOV. Z-stacks for each channel were then collapsed into maximum intensity projections (MIP). To quantify the recruitment of TSG101 and ALIX to the parasite, CellProfiler was used to identify cells (CMDR MIP), nuclei (Hoechst MIP), cell cytoplasm (cell area occluding nucleus and PV) and *Toxoplasma* (GRA1 MIP) and capture intensity measurements of TSG101 or ALIX associated with the PV and cytoplasm to calculate the PV to cytoplasm intensity ratio. All calculations were performed in KNIME. Cells containing multiple PVs were excluded from the analysis.

## Growth analysis of RH, RΔ*gra14* and RΔ*gra14GRA14* complemented strains

To compare the replication of RH, RΔ*gra14* and RΔ*gra14GRA14* complemented strains (RΔ*gra14GRA14*$^{WT}$, RΔ*gra14GRA14*$^{TSG101-}$ RΔ*gra14GRA14*$^{ALIX-}$ and RΔ*gra14GRA14*$^{TSG101-A-LIX-}$) we used the data collected using the Yokogawa CellVoyager 8000 spinning disc confocal high content imaging instrument described above. We measured growth by analyzing the number of parasites per PV determined by the number of DAPI puncta within the PV of GFP-TSG101 infected cells. In addition to that, replication was also assessed by measuring the PV size defined by GRA1 staining.

## Topology of TgGRA14 mutants at the PVM

Analysis of the topology of TgGRA14 encoding late domain motif mutation was performed as previously described [27]. Briefly, HFFs were seeded into chamber slides and infected with RH, RΔ*gra14*, and RΔ*gra14GRA14* complemented strains (RΔ*gra14GRA14*$^{WT}$, RΔ*gra14-GRA14*$^{TSG101-}$ RΔ*gra14GRA14*$^{ALIX-}$ and RΔ*gra14GRA14*$^{TSG101-ALIX-}$) for 24 hpi. The samples were fixed with 4% PFA, followed by blocking for 20 min with 3% bovine serum albumin (BSA). To achieve semi-permeabilization conditions that would leave the PVM intact we used ~0.00001% saponin in 3% BSA for 10 min and stained with antibodies to the TgGRA14 N-terminus and HA (targeting TgGRA14 C-terminus). 0.1% saponin was used for full permeabilization.

## Supporting information

**S1 Fig. Comparison of ALIX and GFP-TSG101 recruitment to the PVM. A.** Representative images for segmentation and quantification of GFP-TSG101 and ALIX to the PVM between

WT, RΔ*gra14* and R:GRA14<sub>OE</sub>. The PV was labeled using an antibody against TgGRA1. Scale bar is 5 μm.
(DOCX)

**S2 Fig. Endogenous C-terminal tagging of TgGRA14 in a type II strain. A.** Schematic for the tagging strategy with primer binding sites. **B.** PCR to validate the tagging of TgGRA14 with HA in the ME49Δ*ku80* background. **C.** Representative images showing the co-localization of HA with TgGRA14 in the tagged strain. Scale bar is 5 μm.
(DOCX)

**S3 Fig. TgGRA14 is not needed for the internalization of host cytosolic proteins early in infection. A.** Experimental design for the analysis of the internalization of host cytosolic proteins of TgGRA14-deficient parasites. (1) Parasites were treated with 5 μM LHVS for 24 h prior to infection, (2) inducible mCherry HeLa cells were infected with parasites for 4 h, (3) the parasites were harvested at 4 hpi and analyzed by microscopy. **B.** Quantification of host cytosolic mCherry uptake at 4 hpi by WT or RΔ*gra14* type I strains treated with DMSO or LHVS for 24 h. At least 200 parasites were analyzed per blinded sample. Data represents the mean from $\geq 3$ biological replicates. Statistical analysis was by Student's t-test. Only statistical differences are shown.
(DOCX)

**S4 Fig. Deletion of TgGRA14 in type II strain. A.** Schematic for the deletion of the HXGPRT gene with primer binding sites **B.** PCR to validate the deletion of HXGPRT. **C.** Schematic for the deletion of the TgGRA14 with primer amplification sites. **D.** PCR to validate the deletion of TgGRA14 in the ME49Δ*ku80* background. **E.** Immunoblot confirming the deletion of TgGRA14 in the MΔ*gra14* strain.
(DOCX)

**S5 Fig. RΔ*gra14:gra14-HA* complementation mutants. A.** Schematic representation of the plasmids generated for TgGRA14 complementation. **B.** PCR to validate the presence of TgGRA14-HA in the RΔ*gra14GRA14* complementation mutants. **C.** Immunoblot confirming the presence of TgGRA4 and HA in the RΔ*gra14GRA14* complementation mutants. **D.** Representative images showing the presence of HA and GRA14 in the RΔ*gra14GRA14* complementation mutants. Scale bar is 5 μm.
(DOCX)

**S6 Fig. Topology of TgGRA14 late domain motif mutants at the PVM. A.** Semi-permeabilization with 0.1% saponin for permeabilization of the host plasma membrane and PVM. **B.** Semi-permeabilization using 0.00001% saponin to only permeabilize the host plasma membrane and not the PVM. Detection of TgGRA14 N-terminus and C-terminus by probing against GRA14N and HA respectively. Scale bar is 5μm.
(DOCX)

**S7 Fig. Comparison of GFP-TSG101 recruitment to the PVM following disruption of late domain motifs. A.** Representative images for segmentation and quantification of GFP-TSG101 to the PVM between WT, RΔ*gra14* and RΔ*gra14* complement strains. The PV was labeled using an antibody against TgGRA1. Scale bar is 5 μm. **B.** Replication analysis of TgGRA14 complement strains in GFP-TSG101 HeLa cells by analyzing parasites (nuclei) per PV (labeled with GRA1). **C.** Measurement of growth by analyzing the PV size of TgGRA14 complement strains in GFP-TSG101 HeLa cells. Data represents the mean from 3 biological replicates. Statistical analysis was by Kruskal-Wallis test. No statistical differences were

detected between samples.
(DOCX)

**S1 Dataset. LC-MS/MS Analysis.** Data are presented in 9 different tabs. The "Results Summary" tab provides an overview and heat map of the top protein hits identified for each of the three different conditions (Tz-HFF: tachyzoite fibroblast infection; Bz-HFF: bradyzoite fibroblast infection; Neuron: neuron infection). The tabs labeled "Condition_PeptideCalc" demonstrate the analysis pipeline, including equations used in Excel to ultimately calculate $\log_2$ protein fold changes and p-values. The tabs labeled "Condition_VolcanoPlot" provide a graphical overview of protein fold changes and short names for the top protein hits. Lastly, the "Peptide_RawData" tabs provide identification scores and PEP values for each unique peptide identified from either Toxoplasma-Human or Toxoplasma-Mouse combined proteome database searches.
(XLSX)

**S1 Table. *T. gondii* proteins encoding a P(S/T)AP motif identified bioinformatically.** Identification of *Toxoplasma gondii* effector proteins encoding a putative P(S/T)AP motif and a signal peptide as potential candidate interactors with the host ESCRT-I component, TSG101. Bioinformatic search was performed using the vEuPathDB *Toxoplasma* Informatics Resources Database (ToxoDB) and their predicted localization is based on the hyperplexed localization of organelle proteins by isotype tagging (hyperLOPIT) data set [36].
(DOCX)

**S2 Table. Host and parasite proteins significantly enriched following immunoprecipitation with TgGRA14-HA.** Immunoprecipitation of TgGRA14 with host proteins. List of TgGRA14 host-interacting proteins and *T. gondii* interacting proteins in Tz-HFF samples with >1 log2 fold change in tagged vs control lysates and a negative log2 p-value >3.32 (equivalent to p<0.1). Comparison of TgGRA14 host-interacting proteins in tachyzoite stage (Tz-HFF) with bradyzoite stage (Bz-HFF) or tachyzoites infected HFF (Tz-HFF) with tachyzoites infected neurons (Tz-Neurons). Cells are color coded with blue indicating high enrichment, white indicating moderate enrichment, and red indicating lower, but still significant, enrichment. N.D., not detected.
(DOCX)

**S3 Table. Parasite strains used in this manuscript.**
(DOCX)

**S4 Table. Primers used in this manuscript.**
(DOCX)

**S5 Table. Plasmids used in this manuscript.**
(DOCX)

**S6 Table. Antibodies used in this study.**
(DOCX)

## Acknowledgments

We thank Dr. Peter Bradley for providing us with the pGRA14-HA plasmid and the RΔ*gra14* and R:GRA14$_{OE}$ parasite strains and Dr. Schuyler B. van Engelenburg for sharing the endogenously tagged GFP-TSG101 HeLa cells with us. We also thank Drs. Yoshifumi Nishikawa, Gary Ward, Furio Spano, L. David Sibley, Jeroen Saeij, Peter Bradley, and Dominique Soldati-

Favre for providing antibodies along with Drs. Phyllis Hanson and Isabelle Coppens for their input for this project.

## Author Contributions

**Conceptualization:** Yolanda Rivera-Cuevas, Vern B. Carruthers.

**Data curation:** Joshua Mayoral, Anna-Lisa E. Lawrence, Einar B. Olafsson.

**Formal analysis:** Yolanda Rivera-Cuevas, Joshua Mayoral.

**Funding acquisition:** Yolanda Rivera-Cuevas, Vern B. Carruthers.

**Investigation:** Yolanda Rivera-Cuevas, Joshua Mayoral, Anna-Lisa E. Lawrence.

**Methodology:** Yolanda Rivera-Cuevas, Joshua Mayoral, Anna-Lisa E. Lawrence.

**Project administration:** Vern B. Carruthers.

**Resources:** Manlio Di Cristina, Dishari Thornhill, Benjamin S. Waldman, Akira Ono, Jonathan Z. Sexton, Sebastian Lourido, Louis M. Weiss.

**Supervision:** Louis M. Weiss, Vern B. Carruthers.

**Visualization:** Yolanda Rivera-Cuevas.

**Writing – original draft:** Yolanda Rivera-Cuevas, Joshua Mayoral.

**Writing – review & editing:** Yolanda Rivera-Cuevas, Joshua Mayoral, Manlio Di Cristina, Anna-Lisa E. Lawrence, Einar B. Olafsson, Romir K. Patel, Dishari Thornhill, Benjamin S. Waldman, Akira Ono, Sebastian Lourido, Louis M. Weiss, Vern B. Carruthers.

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
