## [Decision Letter · Decision Letter 0]

9 Sep 2021

Dear Dr. Carruthers,

Thank you very much for submitting your manuscript "Toxoplasma gondii exploits the host ESCRT machinery for parasite uptake of host cytosolic proteins" for consideration at PLOS Pathogens. As with all papers reviewed by the journal, your manuscript was reviewed by members of the editorial board and by several independent reviewers. In light of the reviews (below this email), we would like to invite the resubmission of a significantly-revised version that takes into account the reviewers' comments.

Thank you for submitting your manuscript “Toxoplasma gondii exploits the host ESCRT machinery for parasite uptake of host cytosolic proteins“ for consideration by Plos Pathogens. Your article has been reviewed by our editors and three peer reviewers. All reviewers felt that your work contains interesting new information and is potentially acceptable for publication in PLOS Pathogens. However, the reviewers did have comments, listed below, that require your consideration in the revision of your manuscript. Please pay particular attention to the following major points raised by the reviewers and give them due consideration:

1. It would be nice to see some example cells for the uptake assays (e.g. Fig. 1F,G; Fig. 6). This is a rather crude assay, wouldn't it be possible to quantify the sum of material taken up per parasite?

2. Can it be excluded that the reduced uptake in the GRA14KO is due to a secondary effect? This might be an alternative explanation for the lack of effect on host cell cytosol uptake when the late domain of GRA14 was mutated.

3. If indeed the host ESCRT is involved in uptake into the parasite, wouldn't one expect to see TSG101 and ALIX in the parasite? Maybe in CPL-KO parasites or after LHVS treatment?

4. In order to support the recruitment of a functional ESCRT machinery at the PVM, it would be important to know whether other components of the ESCRT machinery, like ESCRT-III component CHMP4B, are also present at the PVM. In figure 1C, we would like to see CHMP4B expression (similar to panel B), instead of the PVM marker. Since the ESCRT machinery plays a vital role in membrane remodelling, the authors might also propose a role in deformation of the PVM to promote the formation of the intravacuolar network (IVN), which is known to contribute to parasite uptake (Dou et al. MBio 2014).

5. We wonder if the defect in uptake of host cytosolic proteins in Δgra14 is correlated with a slight defect in growth? The growth rate of Δgra14 mutant is not analyzed in this study and was not properly documented in the previous study of Rome et al (2008) that characterized GRA14 : «the parental strain and the Δgra14 strain parasites grew with approximately the same kinetics (data not shown) ». A replication assay of Δgra14 strain would be informative for this study. Also the authors did not mention in the discussion that the defect in uptake of host cytosolic proteins in Δgra14 parasites is not due to a defect in the formation of IVN, as for GRA2 (Dou et al. MBio 2014), since no noticeable changes in IVN was observed previously in Δgra14 strain (Rome et al. Infect Immun 2008).

6. The determinants of GRA protein topology are unknown. Thus, it is a formal possibility that mutagenesis of the PTAP motif alters the topology of this protein, thereby blocking its function. The paper would be stronger if the topology of this mutant could be assessed.

(1) A letter containing a detailed list of your responses to the review comments and a description of the changes you have made in the manuscript.

(2) Two versions of the manuscript: one with either highlights or tracked changes denoting where the text has been changed; the other a clean version (uploaded as the manuscript file).

We cannot make any decision about publication until we have seen the revised manuscript and your response to the reviewers' comments. Your revised manuscript is also likely to be sent to reviewers for further evaluation.

Sincerely,

Dominique Soldati-Favre

Section Editor

PLOS Pathogens

Dominique Soldati-Favre

Section Editor

PLOS Pathogens

Kasturi Haldar

Editor-in-Chief

PLOS Pathogens

orcid.org/0000-0001-5065-158X

Michael Malim

Editor-in-Chief

PLOS Pathogens

orcid.org/0000-0002-7699-2064

Thank you for submitting your manuscript “Toxoplasma gondii exploits the host ESCRT machinery for parasite uptake of host cytosolic proteins“ for consideration by Plos Pathogens. Your article has been reviewed by our editors and three peer reviewers. All reviewers felt that your work contains interesting new information and is potentially acceptable for publication in PLOS Pathogens. However, the reviewers did have comments, listed below, that require your consideration in the revision of your manuscript. Please pay particular attention to the following major points raised by the reviewers and give them due consideration:

1. It would be nice to see some example cells for the uptake assays (e.g. Fig. 1F,G; Fig. 6). This is a rather crude assay, wouldn't it be possible to quantify the sum of material taken up per parasite?

2. Can it be excluded that the reduced uptake in the GRA14KO is due to a secondary effect? This might be an alternative explanation for the lack of effect on host cell cytosol uptake when the late domain of GRA14 was mutated.

3. If indeed the host ESCRT is involved in uptake into the parasite, wouldn't one expect to see TSG101 and ALIX in the parasite? Maybe in CPL-KO parasites or after LHVS treatment?

4. In order to support the recruitment of a functional ESCRT machinery at the PVM, it would be important to know whether other components of the ESCRT machinery, like ESCRT-III component CHMP4B, are also present at the PVM. In figure 1C, we would like to see CHMP4B expression (similar to panel B), instead of the PVM marker. Since the ESCRT machinery plays a vital role in membrane remodelling, the authors might also propose a role in deformation of the PVM to promote the formation of the intravacuolar network (IVN), which is known to contribute to parasite uptake (Dou et al. MBio 2014).

5. We wonder if the defect in uptake of host cytosolic proteins in Δgra14 is correlated with a slight defect in growth? The growth rate of Δgra14 mutant is not analyzed in this study and was not properly documented in the previous study of Rome et al (2008) that characterized GRA14 : «the parental strain and the Δgra14 strain parasites grew with approximately the same kinetics (data not shown) ». A replication assay of Δgra14 strain would be informative for this study. Also the authors did not mention in the discussion that the defect in uptake of host cytosolic proteins in Δgra14 parasites is not due to a defect in the formation of IVN, as for GRA2 (Dou et al. MBio 2014), since no noticeable changes in IVN was observed previously in Δgra14 strain (Rome et al. Infect Immun 2008).

6. The determinants of GRA protein topology are unknown. Thus, it is a formal possibility that mutagenesis of the PTAP motif alters the topology of this protein, thereby blocking its function. The paper would be stronger if the topology of this mutant could be assessed.

(1) A letter containing a detailed list of your responses to the review comments and a description of the changes you have made in the manuscript.

(2) Two versions of the manuscript: one with either highlights or tracked changes denoting where the text has been changed; the other a clean version (uploaded as the manuscript file).

Reviewer's Responses to Questions

**Part I - Summary**

Reviewer #1: This report shows that the uptake of host cell cytosol into intracellular T. gondii parasites involves the ESCRT machinery of the host cell which is recruited to the outer face of the parasite's PVM. It further identifies GRA14 as a contributor of ESCRT PVM recruitment and dissects the motifs in the GRA14 sequence mediating the recruitment of particular ESCRT components. The role of GRA14 at the PVM is substantiated by its ability to functionally replace a corresponding part in HIV GAG in a cellular virus particle release assay.

This is a very interesting paper providing (i) insight into GRA14 function, (ii) evidence that the host ESCRT machinery contributes to host cell cytosol uptake and (iii) - on a more general scale - that also eukaryotic pathogens exploit the host ESCRT machinery, similar to many viruses and bacteria.

Reviewer #2: In the submitted paper, “Toxoplasma gondii exploits the host ESCRT machinery for parasite uptake of host cytosolic proteins”, the authors discover a role for the ESCRT machinery in host cytosolic protein uptake. Importantly, they demonstrate that uptake is linked to the secreted protein GRA14, whose function has previously been unknown. They additionally show that GRA14 protein regulates association of the ESCRT complex protein TSC101 by a conserved motif in its C-terminal domain. Together, the paper provides new mechanistic insight into how T. gondii recruits the host ESCRT machinery and is an important addition to the field. Specific comments are below, which are mostly minor.

Reviewer #3: In this paper Rivera-Cuevas and colleagues investigated mechanical insights into the parasite uptake of host cytosolic proteins, a process largely unknown in Toxoplasma. They demonstrated for the first time a role for the host ESCRT machinery in this process, but also its role in parasite replication. They found accumulation of ESCRT-I proteins ALIX and TSG101 at the PVM. The authors performed a bioinformatic screen to search for parasite secreted proteins containing ESCRT-I late domains and identified the dense granule GRA14 protein as an effector in the recruitment of TSG101 at the PVM. Several ESCRT proteins and other GRA proteins were enriched in GRA14 immunoprecipitation. The characterization of the Δgra14mutant revealed its contribution in the uptake of host cytosolic proteins, but this does not seem to be linked to its ability to recruit TSG101 at the PVM, leaving the exact role played by GRA14 in this process unanswered.

Overall, this is a nice study that defines both host and parasite players for uptake of host cytosolic proteins. The paper is well written; the experimental approaches and the analyses are well conducted.

**Part II – Major Issues: Key Experiments Required for Acceptance**

Reviewer #1: My only more major concern with this work lies in the host cell cytosol uptake function (which should be easily addressable without further experiments and should not be taken as an obstacle to publish this):

1a. It would be nice to see some example cells for the uptake assays (e.g. Fig. 1F,G; Fig. 6). This is a rather crude assay, wouldn't it be possible to quantify the sum of material taken up per parasite?

1b. Can it be excluded that the reduced uptake in the GRA14KO is due to a secondary effect? This might be an alternative explanation for the lack of effect on host cell cytosol uptake when the late domain of GRA14 was mutated.

1c. If indeed the host ESCRT is involved in uptake into the parasite, wouldn't one expect to see TSG101 and ALIX in the parasite? Maybe in CPL-KO parasites or after LHVS treatment?

1d. Conservation in apicomplex: a host ESCRT-based mechanism for host cell cytosol uptake is not possible in malaria blood stage parasites as their host cells presumably lack ESCRT components. Nevertheless, these parasites efficiently take up host cell cytosol. Do the authors think that the ESCRT-aided mechanism of uptake is specific for T. gondii and fundamentally different to that of malaria blood stage parasites? Or is the ESCRT pathway only one of multiple pathways in T. gondii? Could it be that ESCRT has also other functions at the T. gondii PVM? This might be discussed.

Reviewer #2: (No Response)

Reviewer #3: 1. In order to support the recruitment of a functional ESCRT machinery at the PVM, it would be important to know whether other components of the ESCRT machinery, like ESCRT-III component CHMP4B, are also present at the PVM. In figure 1C, we would like to see CHMP4B expression (similar to panel B), instead of the PVM marker. Since the ESCRT machinery plays a vital role in membrane remodelling, the authors might also propose a role in deformation of the PVM to promote the formation of the intravacuolar network (IVN), which is known to contribute to parasite uptake (Dou et al. MBio 2014).

2. We wonder if the defect in uptake of host cytosolic proteins in Δgra14 is correlated with a slight defect in growth? The growth rate of Δgra14 mutant is not analyzed in this study and was not properly documented in the previous study of Rome et al (2008) that characterized GRA14 : «the parental strain and the Δgra14 strain parasites grew with approximately the same kinetics (data not shown) ». A replication assay of Δgra14 strain would be informative for this study. Also the authors did not mention in the discussion that the defect in uptake of host cytosolic proteins in Δgra14 parasites is not due to a defect in the formation of IVN, as for GRA2 (Dou et al. MBio 2014), since no noticeable changes in IVN was observed previously in Δgra14 strain (Rome et al. Infect Immun 2008).

**Part III – Minor Issues: Editorial and Data Presentation Modifications**

Reviewer #1: Minor points:

FigS1B: Based on the methods it can be assumed that the spinning disk microscopy based quantification is accurate but at least in Fig. S1B it is not so obvious that the GRA14 KO has less TSG101 at the vacuole (the top panel seems to have larger cytoplasmic pool). Are these cells representative? Some explanation how to read these images might be helpful to make this difference more obvious. The same applies to Fig. S6 and Fig. 8.

- Figure 3A. Neither GRA14, ALIX, nor TSG101 really overlap very much. Is this what would be expected if GRA14 binds ALIX when viewed at SIM resolution? The data in Fig. 3B and C clearly support very close proximity and the IP data (Fig. 5A,B) strongly supports an interaction, so there seems little doubt ALIX and GRA14 and TSG101 meet. Do the authors have an explanation for the lack of full overlap in Fig. 3A?

- Legend Figure S2 ' primer amplification sites.' Better 'primer binding site'

- line 277: Remove 'the' in'...within the 30 min post...'

- line 341: remove one of the two 'in'

- some of the references have '[Internet]' after the title, e.g. Ref11. Is this per PLosPath formatting?

Reviewer #2: In the submitted paper, “Toxoplasma gondii exploits the host ESCRT machinery for parasite uptake of host cytosolic proteins”, the authors discover a role for the ESCRT machinery in host cytosolic protein uptake. Importantly, they demonstrate that uptake is linked to the secreted protein GRA14, whose function has previously been unknown. They additionally show that GRA14 protein regulates association of the ESCRT complex protein TSC101 by a conserved motif in its C-terminal domain. Together, the paper provides new mechanistic insight into how T. gondii recruits the host ESCRT machinery and is an important addition to the field. Specific comments are below, which are mostly minor.

1. While the GRA14 P(S/T)AP motif is conserved in Neospora, the YPXL motif does not appear to be. This should be added to the discussion.

Overall, there is little support for this motif in ALIX recruitment (perhaps too close to the PTAP?). At the very least this seems to deserve adding a question mark (or three) to the model in Fig 9?

2. The determinants of GRA protein topology are unknown. Thus, it is a formal possibility that mutagenesis of the PTAP motif alters the topology of this protein, thereby blocking its function. The paper would be stronger if the topology of this mutant could be assessed.

3. Fig. 1C. The authors should clarify how the PVM and Toxo are being detected in this figure.

4. Fig. 1F, G. The authors should clarify what is being scored here. It appears it is the percent of parasites with any visible venus being taken up? Are there differences in the amount of uptake in the wt vs EQ cells? It may be helpful to show a representative field for each here, particularly for this first figure.

5. Fig. 1G. The percentages of replicating parasites that uptake Venus is extremely low (only 6% or so, even in the ∆cpl background). It would help the readers to describe why this is expected in the text.

6. Fig 2D – While the general idea that GRA14 can substitute is clear – the figure is confusing. The quantified data is from multiple experiments. The western does not particularly agree and is not described in the legend)

7. The term GRA14OE is vague – to what extent is this overexpressed?

8. Figure 3A – it is difficult to visualize where the parasites are in this figure. It is also difficult to determine the extent of colocalization as claimed.

9. Fig 3B – the author claim that the PLA assays demonstrate that GRA14 is “intimately associated” with ALIX and TSG101. There are several reports of these assays having larger than the stated 40nm distance (“intimately associated” may be an overstatement).

10. The discussion of the PPYPXXXXYP motif / ALG2 comes out of nowhere. Suggest reworking this to introduce this concept better? (is this exact sequence seen or similar?)

11. Fig 2C. The schematic could be made clearer with text to describe the outcome of the fusion (rather than just color coding).

12. Fig 2D – lines are not aligned well

Reviewer #3: 1. The quantification shows clearly a decrease in the recruitment of GFP-TSG101 in ΔGra14 but this is not visible in the image of Figure 4A and Figure S1. Do the dots of strong intensity remain after depletion of gra14 ? Is the difference visible by eyes or only after quantification?

2. Summary : since there is no demonstration that GRA14 contributes to host cytosolic protein uptake via recruiting the ESCRT machinery, more caution should be taken in the author summary (lanes 54-55.)

3. Fig. 1F : there is not statistic difference in the internalization of host-derived Venus between Vps4A WT and EQ mutant for newly invading parasite (only significant with the untransfected cells). The text deserves more nuancing in the conclusion (lanes 124-126). It is not surprising to see no difference at 30 min post-infection because endocytosis occurs in replicating parasite and GRAs accumulate later during intracellular replication. This result also correlates with the fact that GRA14 mutant showed normal ingestion at early time point.

4. Fig. 3. Are all the vacuoles decorated with ALIX and TSG101, or only a subset? What is the % of vacuoles with an accumulation at the PVM?

5. Fig. 1D : the figure captions read «+Δcpl mutant » while the legend reads «infection by RH strain».

6. Figure 4A: the figure legend does not correspond to figure 4A, but instead to figure S1. It would be much better to have figure S1 as main figure, instead of Figure 4A.

7. It is not clear in the text, figure and materials and methods, if the complementation of GRA14 forms are under the gra14 promoter or if they are over expression.

8. It is not mentioned in the legend of figures which antibodies is used to stain the PVM (Figure1C, Fig. 8A ; Fig. S1, Fig. S6).

9. Lane 123 : For readers non expert in the traffic of Toxoplasma, add a sentence to explain the reason to use Δcpl mutant or inhibitor in the experiments of host cytosolic proteins uptake.

10. Fig. 2C : the schematic for domain substitution between Gag and GRA14 is too succinct. Show the exact length and sequence of the swapping.

11. The sizes of the amplification PCR and molecular weight are missing in all figures

12. Line 574. The verb in the sentence is missing

13. Lines 408, 411 : TgGRA14 gene is not italic.

14. The authors should also mention in discussion a recent study on the role of GRA14 in the modification and control of the immune Response Mediated via the NF-κB Pathway (Ihara et al. Frontiers in Immunology 2020).

PLOS authors have the option to publish the peer review history of their article (what does this mean?). If published, this will include your full peer review and any attached files.

Reviewer #1: No

Reviewer #2: No

Reviewer #3: No
---

## [Editor Report · Decision Letter 1]

23 Nov 2021

Dear Dr. Carruthers,

We are pleased to inform you that your manuscript 'Toxoplasma gondii exploits the host ESCRT machinery for parasite uptake of host cytosolic proteins' has been provisionally accepted for publication in PLOS Pathogens.

Best regards,

Philipp Olias

Associate Editor

PLOS Pathogens

Dominique Soldati-Favre

Section Editor

PLOS Pathogens

Kasturi Haldar

Editor-in-Chief

PLOS Pathogens

orcid.org/0000-0001-5065-158X

Michael Malim

Editor-in-Chief

PLOS Pathogens

orcid.org/0000-0002-7699-2064
---

## [Editor Report · Acceptance letter]

7 Dec 2021

Dear Dr. Carruthers,

We are delighted to inform you that your manuscript, "*Toxoplasma gondii* exploits the host ESCRT machinery for parasite uptake of host cytosolic proteins," has been formally accepted for publication in PLOS Pathogens.

Best regards,

Kasturi Haldar

Editor-in-Chief

PLOS Pathogens

orcid.org/0000-0001-5065-158X

Michael Malim

Editor-in-Chief

PLOS Pathogens

orcid.org/0000-0002-7699-2064